# Perineuronal nets stabilize the grid cell network

Ane Charlotte Christensen [1,2,3], Kristian Kinden Lensjø [2,3,5], Mikkel Elle Lepperød [1,2,3,5],
Svenn-Arne Dragly [3,4], Halvard Sutterud[3,4], Jan Sigurd Blackstad [2,3], Marianne Fyhn[2,3] &
Torkel Hafting [1,3 ✉]

Grid cells are part of a widespread network which supports navigation and spatial memory.
Stable grid patterns appear late in development, in concert with extracellular matrix aggregates termed perineuronal nets (PNNs) that condense around inhibitory neurons. It has been suggested that PNNs stabilize synaptic connections and long-term memories, but their role in the grid cell network remains elusive. We show that removal of PNNs leads to lower inhibitory spiking activity, and reduces grid cells' ability to create stable representations of a novel environment. Furthermore, in animals with disrupted PNNs, exposure to a novel arena corrupted the spatiotemporal relationships within grid cell modules, and the stored representations of a familiar arena. Finally, we show that PNN removal in entorhinal cortex distorted spatial representations in downstream hippocampal neurons. Together this work suggests that PNNs provide a key stabilizing element for the grid cell network.

[1] Institute of Basic Medical Sciences, University of Oslo, Oslo, Norway. [2] Department of Biosciences, University of Oslo, Oslo, Norway. [3] Center for Integrative Neuroplasticity, University of Oslo, Oslo, Norway. [4] Department of Physics, University of Oslo, Oslo, Norway. [5] These authors contributed equally: Kristian Kinden Lensjø, Mikkel Elle Lepperød. ✉email: torkel.hafting@medisin.uio.no

The ability to encode novel environments without compromising old memories are vital for survival and cognitive functions. Spatially tuned neurons in the hippocampus and entorhinal cortex are key units for navigation and spatial memory. Neurons in the medial entorhinal cortex (MEC) represent information such as self-location[1], the direction of the animal's head[2], proximity to geometric borders[3], speed[4], and possibly the distance traveled by the animal[5]. The precise position of the animal can be decoded from the activity of grid cell ensembles[1,6], where each cell has multiple firing fields forming a characteristic hexagonal pattern spanning the entire surface of the area visited by the animal[7].

Grid cells provide input to the hippocampus[8] and are assumed to be the primary determinant of hippocampal place cell firing[3,5,9]. However, this notion has been challenged by the fact that place cells appear before grid cells during development[10,11]. In rodents, grid cell firing patterns emerge around postnatal days 16–18 (P16–P18) and transition over time from being unstable and non-periodic to highly regular, reaching adult level grid scores around P28–P34[10,12]. Once established, the periodic spiking pattern of grid cells is remarkably stable when animals revisit the same environment. Local perturbations of different cell types in MEC cause changes to grid field spike rates or an increased number of out-of-field spikes, but the location of the fields remains unaltered[13–16]. Furthermore, when placed in a different environment, neighboring populations of grid cells show a coherent shift of grid fields[17] that retains their relative spatiotemporal relationships[18]. In contrast, the hippocampal place cells remap by independently and unpredictably changing their activity. This suggests that the mature grid cell network is more hardwired and less plastic than the hippocampal place cell network.

The network that controls grid cell spiking is not yet fully understood, but recurrent inhibition appears to be fundamental to the specific activity of grid cells. Stellate cells, one of the two principal cell types that display grid cell firing[8], are connected via parvalbumin expressing (PV$^+$) inhibitory interneurons[19,20]. PV$^+$ cells account for about 50% of the inhibitory cells in MEC[21], making them the main inhibitory mediator in the local grid cell network. In sensory cortex, PV$^+$ cells play a central role in shaping the excitatory activity of principal neurons by regulating the onset of periods of high synaptic plasticity[22]. Whether PV$^+$ inhibitory neurons in MEC play a similar role for development of the grid cell network remains elusive.

A hallmark of maturing PV$^+$ cells is that aggregates of specialized extracellular matrix, called perineuronal nets (PNNs), condense on the cell soma and proximal dendrites, leaving openings only for synaptic connections[23]. PNNs are believed to help stabilize the activity of PV$^+$ cells by supporting synaptic integrity and limiting synaptogenesis, in addition to supporting PV$^+$ cell physiology[24]. By the time PNNs in sensory cortex are fully mature, plasticity in the local network is strongly reduced. However, juvenile levels of plasticity can be reinstated by experimentally removing PNNs in adult animals, which both increases structural plasticity and reduces inhibitory spiking[25–27]. Interestingly, the timeline for maturation of PNNs in MEC coincides with the timeline for the development of grid cell firing[12,28]. This co-occurrence suggests that grid cell activity could be shaped during the developmental period with high levels of plasticity, and that the later presence of PNNs ensures stability of established synaptic connections, and thus maintains the integrity of the network and the spatiotemporal relationship between grid cells.

To test if PNNs support the stability of the grid cell network, we experimentally disrupted PNNs in MEC of adult rats and recorded from single units while animals explored a familiar arena or a novel environment. We observed reduced inhibitory spiking activity when PNNs were removed, and grid cells displayed reduced spatial specificity and spatial information in the familiar environment. We replicated these findings in a simulated network with altered synaptic weights that gave rise to reduced inhibition. This is in line with the role for inhibitory neurons in shaping grid cell activity. When the MEC network was challenged with new information by letting animals explore a novel environment, both the spatial correlations of grid cells and their pairwise temporal correlations were significantly reduced in animals lacking PNNs. This indicates that the novel place code remained unstable and the spatiotemporal relationship between grid cells was impaired. The exposure to a novel environment also destabilized the subsequent representation of the familiar arena, suggesting that PNNs are important for maintaining consistent grid cell representations when a previous environment is revisited.

Finally, we recorded place cells from hippocampal area CA1 when PNNs were removed in MEC. Our data show that the local changes we observed in MEC were also reflected in place cell coding, supporting the idea that the stability and high spatial specificity of grid cell representations are necessary to provide accurate spatial information to place cells.

Together, our data show that the presence of PNNs ensures precise spatial and temporal coding needed to maintain the network configuration of grid cell and place cell circuits.

## Results

**Degradation of PNNs in MEC.** The dense expression of PNNs in MEC of adult rats shows an almost complete overlap with expression of PV and mainly enwraps PV$^+$ cell soma and proximal dendrites (Fig. 1a). We used local injections of the bacterial enzyme Chondroitinase ABC (chABC) to degrade PNNs, and verified the efficiency in histology sections by labeling of *Wisteria floribunda* agglutinin (WFA)-positive PNNs and 6-sulfated unsaturated disaccharides (3B3 "stubs") that are left after the degradation process. Staining for WFA and 3B3 stubs clearly delineated the area affected by chABC (Fig. 1b and Supplementary Fig. 2).

PNNs are hypothesized to limit plasticity by stabilizing synaptic connections and facilitating the high spiking activity of PV$^+$ cells[29–31]. To test how PNNs affect synaptic stability in MEC, we first degraded PNNs unilaterally using chABC. Five days after chABC treatment, we quantified synaptic boutons contacting the soma of PV$^+$ cells via immunostaining. Because environmental enrichment such as exploration of novel arenas or changes to the housing environment can affect neural plasticity[32], animals were kept in their home cage throughout the experiment to isolate the effect of PNN degradation.

In the control hemisphere, we found that a large fraction of inputs to PV$^+$ cell somas were positive for the inhibitory presynaptic marker VGAT (Fig. 1c). Out of the labeled excitatory connections onto PV$^+$ cells, VGLUT1-positive puncta dominated, while a minority were positive for VGLUT2 (Supplementary Fig. 1). Removing the PNNs caused a reduction in the number of VGAT expressing puncta onto PV$^+$ cells ($p = 0.002$, Mann–Whitney $U$ test, $N = 3$ rats, $n$ aCSF = 31 cells, $n$ chABC = 32 cells). No significant changes were observed for excitatory VGLUT1 and 2 puncta (Fig. 1d), although there was a tendency for reduction in VGLUT1 puncta ($p = 0.078$). To investigate the functional role of PNNs, we next recorded single unit activity from MEC using bilaterally implanted tetrodes in 13 rats (7 of which were treated with chABC) while the animals explored a familiar open field arena (Fig. 1e). All units recorded in a familiar environment were separated into narrow- and broad spiking based on waveform properties (Supplementary Fig. 3)[33,34]. Narrow spiking units are putative inhibitory neurons of which PV$^+$ cells constitute the largest group[21,35], while broad

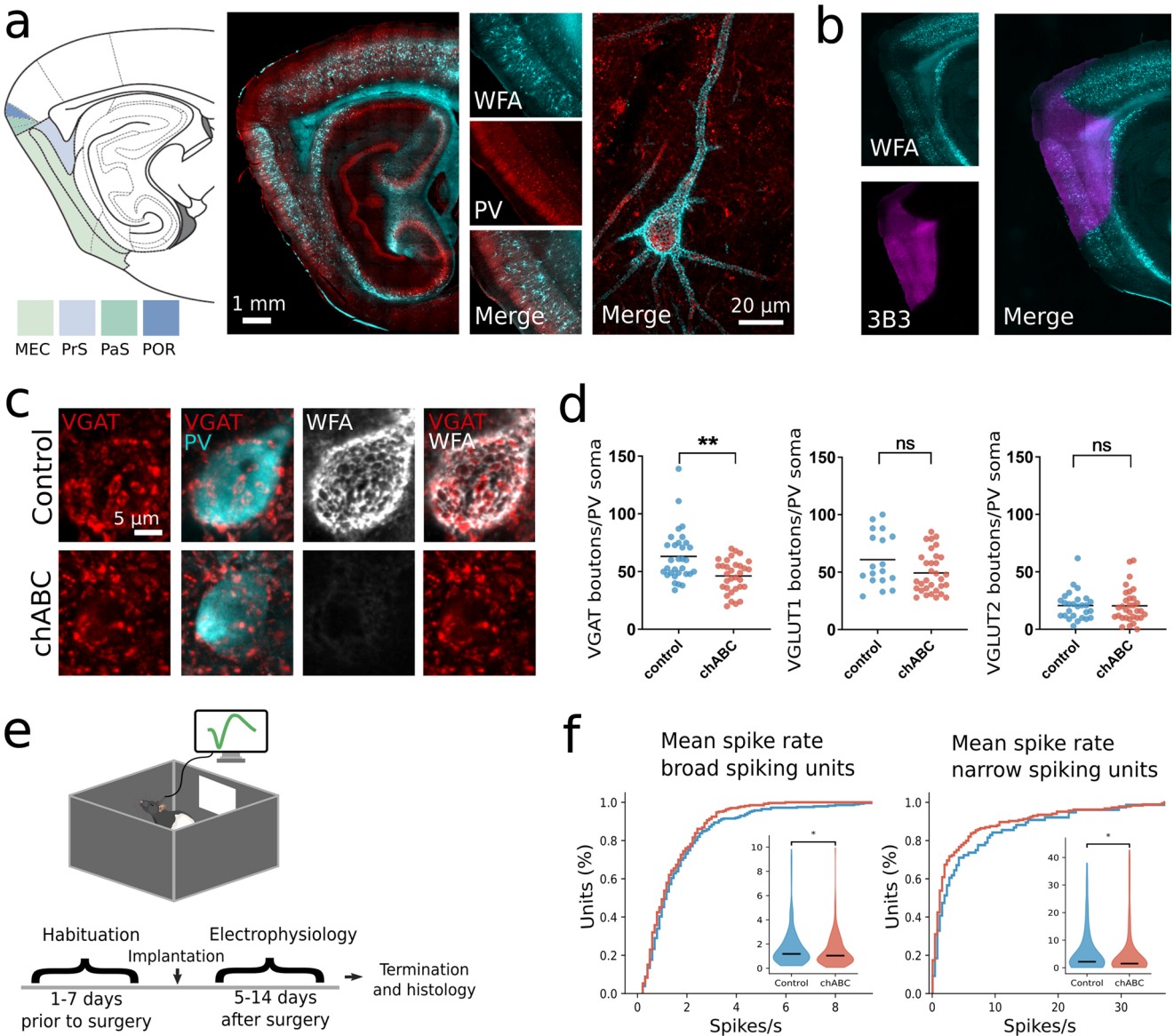

**Fig. 1 Degradation of perineuronal nets alters synaptic connections and neuron spiking activity. a** Schematic outline of parahippocampal areas. Sagittal section from rat brain stained for WFA+ PNNs (cyan) and PV+ neurons (red). PNNs are strongly expressed in MEC and parasubiculum (PaS), while there is weaker staining in presubiculum (PrS) and postrhinal cortex (POR). The overlap between PNNs and PV+ neurons is high. Right, high-magnification image of PV+ neuron in MEC, note that PNN enwrap the cell soma and large parts of proximal dendrites. **b** Histological verification of chABC activity. Anti-chondroitin sulfate (3B3) antibody (magenta) label 6-sulfated unsaturated disaccharide (C-6-S) stubs left from enzymatic degradation of PNNs by chABC. Intact PNN staining (cyan) is greatly reduced after chABC injection and C-6-S stubs shows the full extent of the injection area. **c** VGAT expressing puncta on a PV+ cell in MEC 5 days after injection of aCSF (control, top panel) or chABC (lower panel). **d** The number of VGAT expressing puncta on PV soma is reduced after local chABC treatment in MEC (VGAT: mean ± s.e.m.; Control ($n = 31$ cells) 63.29 ± 4.038; chABC ($n = 32$) 46.34 ± 2.48, $p = 0.002$; VGLUT1: Control ($n = 18$) 60.94 ± 5.48; chABC ($n = 32$) 49.44 ± 3.13, $p = 0.078$; VGLUT2: Control ($n = 27$) 20.54 ± 2.64; chABC ($n = 32$) 20.74 ± 2.35, $p = 0.67$. $N = 3$ animals, Mann–Whitney $U$ test (two sided). **e** Illustration of open field recording setup and timeline for experiments. **f** Cumulative distribution of mean spike rates for broad- and narrow-spiking units from controls and chABC-treated rats. Both unit types showed reduced spike rates in animals with disrupted PNNs (median ($n$); control broad spiking 1.16 Hz (296); chABC broad spiking 1.01 Hz (287), $p = 0.01$, $U = 47688$; control narrow spiking 2.46 Hz (73); chABC narrow spiking 1.52 Hz (177), $p = 0.03$, $U = 7586$, Mann–Whitney $U$ test (two sided). Insets: violin plot shows min to max and median (large black line). ns = not significant, *$p < 0.05$, **$p < 0.01$. Source data are provided as a Source Data file.

spiking units are putative excitatory neurons. A total of 283 units were identified as broad spiking in the control group and 278 in the chABC group. The number of narrow-spiking units were 76 (20%) in the control group and 187 (40%) in the chABC group. Both broad- and narrow-spiking units showed decreased mean spike rates in chABC-treated animals (Fig. 1f and Supplementary Fig. 3). These results are in line with recordings from the visual cortex[26].

**Grid cell representations in familiar environments**. Grid cells are widely thought to correspond to layer II stellate cells in MEC[13,36]. These stellate cells are mainly interconnected with PV+ neurons, contributing to the sharp receptive fields and low out-of-field noise of grid cells[13,19,20,37]. To examine if PNNs are required for grid cell spiking activity, we identified grid cells from all familiar environment recordings using a dual criteria based on gridness score and spatial information analysis. Units

that had scores above the 95th percentile for both gridness and spatial information, generated from shuffling each units' own spikes, were classified as grid cells. The cells were recorded over multiple sessions for several days while the tetrodes were adjusted downwards between sessions to maximize the number of units sampled from each animal. Thus, if a unit was recorded more than once, only the first recording was included in statistical analysis of grid cell spiking properties (Supplementary Table 1). From this, we identified a total of 840 unique units, of which 23% displayed grid cell spiking activity in the control group (86 out of 373 units), and 14% in chABC-treated animals (63 out of 467 units).

In animals treated with chABC we observed more dispersed spiking within and at the edges of grid fields (Fig. 2a, b), but this did not lead to reduction in gridness scores (Fig. 2c). Pharmacogenetically reducing activity in PV$^+$ neurons lead to increased out-of-field activity in grid cells[14]. To test if similar effects could be seen after removal of PNNs, we identified firing fields for each grid cell (Supplementary Fig. 4) and calculated mean firing rate inside and outside of fields. Grid cells from chABC-treated rats showed a weak but consistent tendency of reduced firing rates inside grid fields and increased firing rates outside grid fields (Supplementary Table 2). The combination of these effects lead to a prominent decrease in spatial specificity of grid cells in animals with disrupted PNNs (Fig. 2d). To account for effects of an arbitrary definition of grid fields, we also calculated the spatial information content of each grid cell, a measure that is independent of any predefined firing fields[38]. The spatial information was significantly reduced in grid cells from animals treated with chABC (Fig. 2e). In addition, we found changes in the temporal spiking properties of grid cells. The maximum firing rate was significantly reduced (Fig. 2f), along with the fraction of bursting events (Fig. 2g). This was accompanied by a reduction in spiking variability, measured by the coefficient of variation (CV) of interspike intervals (ISI) (Fig. 2h and Supplementary Fig. 5).

As animals move through the environment, grid cells are thought to be updated by path integration through cells that monitor the animal's instantaneous speed[4]. The impairment in

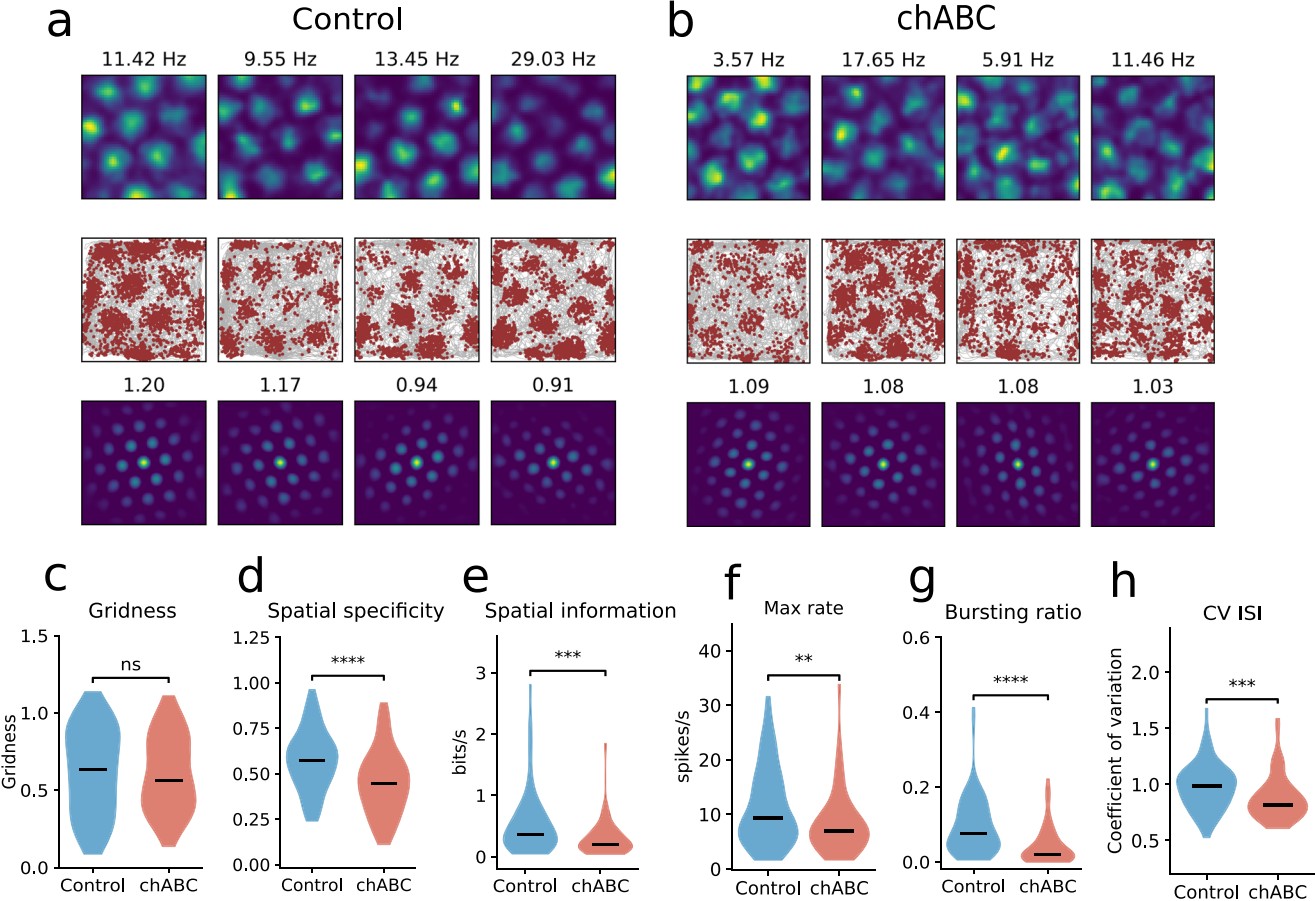

**Fig. 2 Grid cells show reduced spatial coding after PNN removal. a**, **b** Example grid cells from control and chABC-treated rats. Color-coded rate maps (top), running path with spikes superimposed (middle) and spatial autocorrelation maps (bottom) for four grid cells in controls (**a**) and four grid cells in chABC-treated rats (**b**). The number above the rate map is maximum firing rate and the number above the autocorrelation map is gridness score. **c** Removing PNNs did not reduce gridness in the familiar environment (median; Control 0.63; chABC 0.56, p = 0.694). **d** The dispersed spiking outside grid cell fields caused the spatial specificity of grid cells to decrease (median; Control 0.57; chABC 0.45, p < 0.0001), in addition to reducing the **e** spatial information (median; Control 0.36; chABC 0.2, p < 0.0001). **f** Maximum firing rate was reduced (median; Control 9.32; chABC 6.93, p = 0.009). **g** Grid cell bursting ratio was measured as the number of bursting events divided by the number of single-spike events. Rats with disrupted PNNs showed a large reduction in bursting event ratio (median; Control 0.08; chABC 0.02, p < 0.0001). **h** Spiking variability was measured as the coefficient of variation (CV) of interspike intervals (ISI). The method used here calculates CV of ISI using only spikes from passes though grid fields (Supplementary Fig. 5) (median; Control 0.99; chABC 0.82, p < 0.001). Violin plots show min to max and median (black line). Width of graph corresponds to number of samples for each value. Control n = 86, chABC n = 63. Mann–Whitney U test (two sided). **p < 0.01, ***p < 0.001, ****p < 0.0001. Source data are provided as a Source Data file.

the spatial selectivity of the grid cells after PNN removal might reflect impaired integration of speed information. To test if speed modulation was affected by PNN removal, we identified cells that had a speed score above the 95th percentile of shuffled spikes ($n = 323$ for control, $n = 300$ for chABC). The speed score for all recorded units was significantly higher in chABC-treated rats in the familiar environment. When only comparing pure "speed cells" (see "Methods"), we did not observe any change in speed score (Supplementary Fig. 6 and Supplementary Table 9).

It remains unclear whether $PV^+$ cells are necessary for directional tuning. Previously it has been shown that $PV^+$ interneurons inhibit firing in head direction cells[13]. Removal of PNNs reduced the head direction score (vector length) and the maximal firing rate was lower than in the control group (Supplementary Fig. 7). Similar to grid cells, both head direction cells and speed cells showed a significant reduction in bursting and spiking variability in the familiar environment (Supplementary Table 9 and Supplementary Table 10). Taken together this suggests that the effect of PNN degradation goes beyond the effect of reducing the spiking activity of $PV^+$ cells.

**Impaired grid cell representations in novel environment**. Previous work from other cortical areas has demonstrated that PNN removal dramatically increases neural plasticity. However, the removal does not cause major disturbances to normal network function unless the system is required to encode new information. In the visual cortex, properties such as receptive fields and visual acuity are intact when the PNN is removed, but undergo dramatic changes if vision is occluded from one eye[25,26]. Since we observed only subtle effects on grid cell spiking properties in the familiar environment when PNNs were removed, we wanted to test how the grid cell network responded to encoding new information. We introduced the animals to a novel environment which cause grid cells to remap (shift their grid map)[17], expand and become spatially unstable[39], an effect that is reversed when animals explore the environment to the point where it becomes familiar.

When multiple stable grid cells could be recorded from an animal in the familiar environment, we started a novel environment experiment that consisted of a familiar environment recording session (named Familiar I), followed by three sessions in a similar box in a different room (Novel I, II, and III) and a final session in the familiar room (Familiar II). All sessions lasted 20 min and were separated by a 5–10 min break where the animal rested in their home cage (Fig. 3a).

While the grid cells' gridness scores was similar between groups in the familiar environment, introduction to a novel environment lead to a reduction in gridness scores in animals with disrupted PNNs. When averaging all three sessions in the novel environment, control animals showed gridness scores similar to the familiar environment, while the chABC-treated group had substantially reduced gridness scores in the novel versus familiar environment (Fig. 3b and Supplementary Table 3). We then tested how fast newly formed grid maps adapted a stable spatial spiking pattern by correlating rate maps for every five minutes of the first two novel environment recording sessions (Novel I and II) with the rate map from the last 20 min in the novel room (Novel III) (spatial correlation). In the novel environment experiments, only units that were categorized as grid cells in familiar I and could be followed throughout all recording sessions were included in this analysis. For this analysis we identified 16 grid cells from control animals and 16 grid cells from chABC-injected animals. Animals injected with chABC in MEC showed significantly lower spatial correlations in the novel environment (main effect of group: $F_{(7, 210)} = 4.905$, $p < 0.0001$; interaction: $F_{(7, 21)} = 4.91$, $p < 0.0001$, Control $n = 16$, chABC

$n = 16$, two-way repeated measures ANOVA with group and time as factors). Correlations were lower in all but the first two periods, 0 to 5 min and from 5 to 10 minutes (Šidák's multiple comparisons post hoc test, Supplementary Table 8). Animals treated with chABC did not reach spatial correlation levels comparable to control animals within 60 min of exploring the novel environment (Fig. 3c).

We next aimed to test the spatial stability of the grid maps during each session by calculating the within-trail spatial correlation. To calculate spatial stability, the rate map from the first 10 min of a recording session was correlated against the rate map from the last 10 min of the session. Within-trail stability was significantly reduced from the familiar environment to the novel environment in the chABC group but not in the control group, and remained reduced throughout all sessions in the novel environment (Fig. 3d). Interestingly, while the spatial stability in the first familiar session did not differ between the two groups, the brief exposure to the novel arena affected the subsequent coding of the familiar environment in chABC-treated animals, as shown by reduced spatial stability within the familiar II trial (Fig. 3e). Importantly, we did not see any difference between groups in behaviors, such as running speed or distance traveled in either the familiar or novel environments (Supplementary Fig. 8). It has been suggested that expansion of the grid scale in a novel environment serves as a novelty signal[39]. We observed grid expansion in both control and chABC-treated animals and the grid spacing returned to smaller spatial scales as the environment became familiar (Supplementary Fig. 9). To assess the long-term stability of novel environment spatial representations, we repeated the novel environment experiment protocol for three consecutive days (on days 2 and 3 we only conducted one 20 min session in the novel environment) (Fig. 3f). From the 16 grid cells we had in each group on day 1, we were able to identify 12 (control) and 11 (chABC) as the same units on day 2, and 12 (control) and 5 (chABC) on day 3. These units were used for comparing spatial correlation from day 1 to day 2 and from day 2 to day 3. Animals treated with chABC continued to show reduced spatial correlation when comparing the last novel environment recording session on day 1 (Novel III) with the novel environment recording session on day 2. This indicates that novel spatial maps were not properly stabilized during the 60 min exploration time on day 1. There was also a tendency towards reduced spatial correlation in the novel environment from day 2 to day 3, but the units identified as the same from day 1 to day 3 in the chABC-treated group were too few to ensure proper statistical power (Fig. 3g).

**PNN removal alters theta oscillations**. Local field potentials (LFPs) in MEC show strong oscillations in the theta frequency range (6–12 Hz) when rats are moving. These oscillations modulate grid cell spiking and organize neuronal activity into temporal windows that is believed to be important for episodic memory formation and plasticity[40,41]. $PV^+$ neurons are vital for producing synchronized activity in local networks and for organizing information transfer between brain areas[42,43]. Hence, changes in $PV^+$ cell function and inhibitory plasticity could have a widespread effect on the synchronized activity of the MEC network. To address this we analyzed the LFPs recorded from MEC of 14 animals (of which 7 were injected with chABC) (Supplementary Fig. 10). We found that LFPs were altered in the theta range of chABC-treated rats (Fig. 4a). The average power of theta oscillations was stronger and the peak frequency decreased in the chABC-treated rats in both novel and familiar environments (Supplementary Table 4).

For both groups the first introduction to the novel environment (Novel I) reduced the theta peak frequency in the power

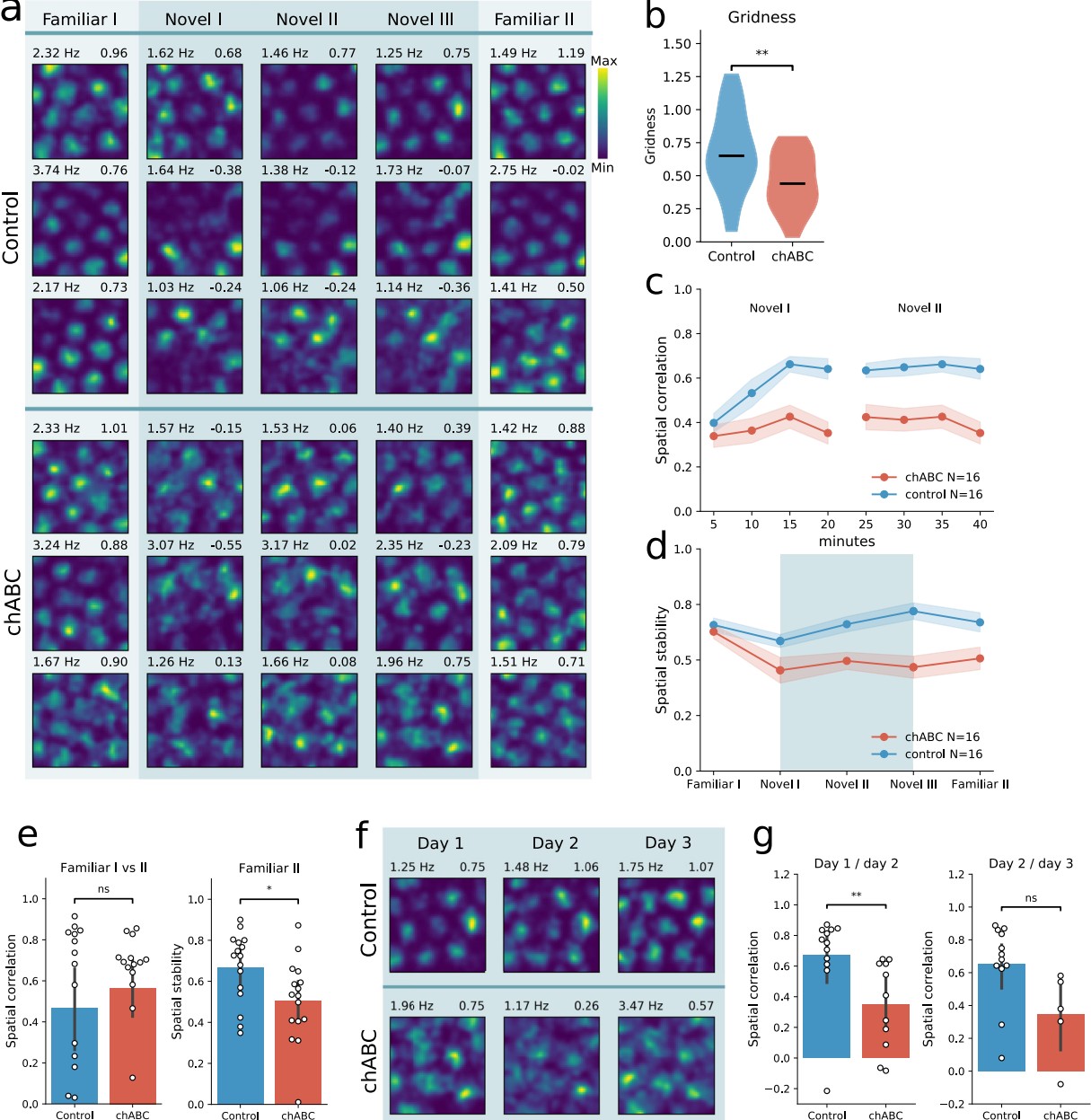

**Fig. 3 Reduced grid cell stability in novel environment. a** Rate maps of five consecutive 20 min recording sessions from six individual grid cells, recorded in control animals (top) and chABC animals (bottom) during a novel environment experiment. Color code indicates spike rate. Rows correspond to individual units and columns correspond to environment. Maximum spike rate (left) and gridness (right) is denoted above each rate map. **b** Gridness score is reduced in the novel environment in animals treated with chABC (median (*n*); Control 0.65 (35); chABC 0.44 (24); *p* = 0.004, *U* = 606, Mann–Whitney *U* test (two sided)). **c** Spatial correlation measured for blocks of 5-min recordings in the novel environment sessions I and II, measured against the novel environment session III. Only units reaching gridness threshold in Familiar I and that could be identified in all five recording sessions were included. Animals treated with chABC showed reduced spatial correlations in the novel environment (mean ± s.e.m.; Control 0.59 ± 0.03; chABC 0.37 ± 0.04; main effect of group: *F*(1, 30) = 14.22, *p* = 0.0007; interaction: *F*(7, 210) = 4.905, *p* < 0.0001, Control *n* = 16; chABC *n* = 16, two-way repeated measures ANOVA with group and time as factors). Points indicate mean; shaded area represents s.e.m. **d** Spatial stability of the same units as in **c**, measured by the within-trial spatial correlation of grid cell rate maps (0–10 min correlated with 10–20 min). Points indicate mean; shaded area represents s.e.m. Spatial stability decreased from Familiar I to Novel I in the chABC-treated group, but not in the control group (median; Control Familiar I 0.67; Control Novel I 0.54, *p* = 0.086; chABC Familiar I 0.65; chABC Novel I 0.45, *p* = 0.015, Mann–Whitney *U* test (two sided)). **e** Spatial stability was similar for both groups in Familiar I (median; Control 0.67; chABC 0.65, *p* = 0.629). After exploring the novel environment, chABC animals showed reduced spatial stability upon returning to the familiar environment (Familiar II) (median; Control 0.72; chABC 0.53, *p* = 0.018, Control *n* = 16; chABC *n* = 16, Mann–Whitney *U* test). **f** Rate maps of two grid cells when introduced to the novel environment for three consecutive days. **g** Spatial correlation of grid cells from chABC-treated animals was decreased in the novel environment when comparing day 1 with day 2 (median; Control 0.76, *n* = 12; chABC 0.41, *n* = 11, *p* = 0.004, Mann–Whitney *U* test (two sided)). Rate map from the third novel session (Novel III) on the first day was used for correlations with day 2. n.s. = not significant, \**p* < 0.05, \*\**p* < 0.01. Source data are provided as a Source Data file.

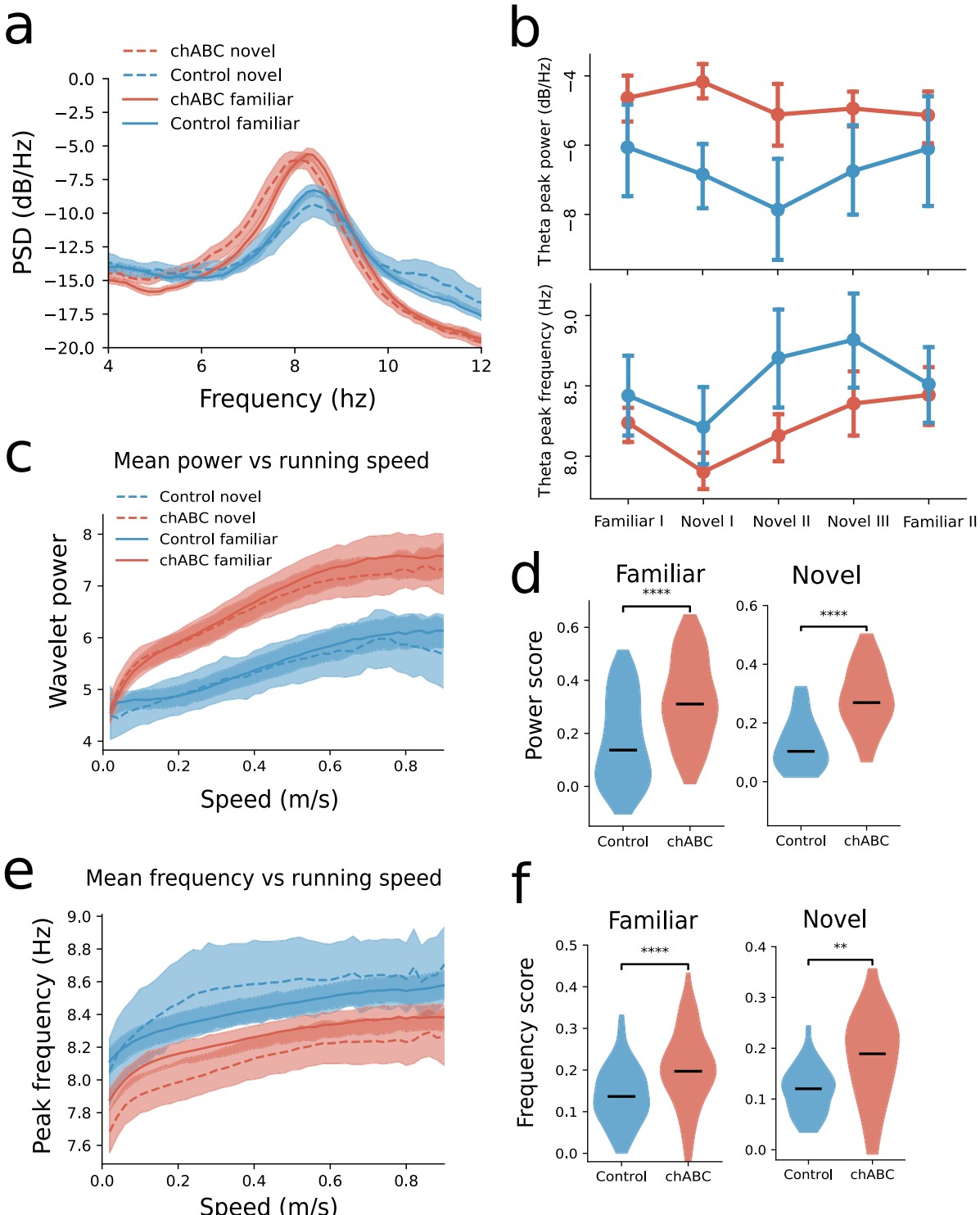

**Fig. 4 Changes in theta oscillations after PNN removal. a** Power spectrum of local field potential in MEC shows that theta oscillations in chABC rats have increased power (dB/Hz) (median; Control familiar 6.12; chABC familiar 4.18; Control novel 7.17; chABC novel 4.8) and lower peak frequency (median; Control familiar 8.39; chABC familiar 8.27; Control novel 8.39; chABC novel 8.09) (solid line: familiar, dotted line: novel, blue: control, red: chABC). Shaded area represents 95% confidence interval. **b** Mean theta frequency for all animals during recordings in familiar and novel environments (dark colored line). Note the decrease in theta frequency during first exposure to a novel environment in both groups. Dots indicate mean; error bars represent bootstrapped 95% confidence interval. **c** The mean theta power was similar when the animal was immobile, but the chABC group showed increased theta power during movement. **d** Power score shows increased correlation between theta power and running speed in animals treated with chABC for both familiar and novel environments (median familiar; Control 0.14 ($n = 148$ recording sessions); chABC 0.31($n = 75$) ($U = 2800$, $p < 0.0001$). Novel; Control 0.10 ($n = 33$); chABC 0.27 ($n = 24$) ($U = 102$, $p < 0.0001$)). **e** Theta frequency increased slightly with running speed in both groups, but the peak frequency was lower in the chABC-treated group. **f** Frequency score shows increased correlation between theta frequency and running speed in chABC-treated animals for both familiar and novel environment (median familiar; Control 0.14 ($n = 148$); chABC 0.20 ($n = 75$) ($U = 3100$, $p < 0.0001$). Novel; Control 0.12 ($n = 33$); chABC 0.19 ($n = 24$) ($U = 200$, $p = 0.002$)). Number of animals: Control $n = 7$, chABC $n = 7$. Mann–Whitney $U$ test (two sided) \*\*$p < 0.01$, \*\*\*\*$p < 0.0001$. Source data are provided as a Source Data file.

spectral density (PSD) (Fig. 4b), while continued exploration of the novel environment (Novel II and III) lead to a large increase of theta frequency in the control group and a smaller increase in the chABC group.

Theta power and frequency is known to increase with the animals' running speed[44,45]. Hence, to test if the difference in theta between the groups was caused by differences in the animals' running speed, we measured running speed and calculated a continuous wavelet transform, conditioned average theta power and peak frequency in 1 s time bins of every session. As expected, the peak theta power and frequency had a positive correlation with running speed for both groups (Fig. 4c, e). We found increased theta power in chABC animals for all running speeds in both familiar and novel environments. The only exception was during immobility where the theta power between groups was similar (Fig. 4c). Correspondingly, the frequency of theta was reduced in chABC-treated animals relative to controls, regardless of running speed (Fig. 4e). To assess the differences in the correlations between power and running speed we calculated a power score. (Fig. 4d). This was significantly stronger for chABC-treated animals in both familiar and novel environments, despite similar running speeds (Supplementary Fig. 8a, b). We did the same analysis for theta frequency and running speed (frequency score), where chABC-treated animals again showed a significantly higher frequency score (Fig. 4f). Hence, both theta power and theta frequency were more correlated with running speed in animals lacking PNNs, meaning that neither the altered theta frequency nor power is well explained by running speed.

**Reduced pairwise grid cell stability in a novel environment**. Grid cells that share similar spacing and orientation maintain their relative spatial relationship when the grid map changes in a novel environment[17,18,46], and the extent of temporal pairwise correlation reflects the degree of spatial overlap. This is indicative of a highly structured and fixed network. Our results suggest that PNN removal increases the potential for plasticity and destabilizes the network in a novel environment. We therefore tested whether this destabilization affected relative spatial and temporal properties within the grid cell population.

We calculated the pairwise spatial and temporal cross-correlation function of grid cells recorded within each experimental paradigm (Familiar I and II and Novel I, II, and III). We next calculated the correlation of the cross-correlation functions across respective paradigms (for example Familiar I vs Novel I)— if the pairwise spatiotemporal relationship is constant, this correlation should be high. Since we already observed changes in LFP theta, we controlled for theta correlations by removing the band (4–10 Hz) from the firing rate of each neuron.

Control animals showed strong correlations of temporal cross-correlations between the familiar and the novel environment (Fig. 5a, c). However, animals treated with chABC showed reductions in pairwise correlations when introduced to a novel environment (Fig. 5b, c), and were significantly less correlated than controls (Fig. 5d). This was also seen when correlating spatial fields (Fig. 5d and Supplementary Fig. 11). Interestingly, chABC animals also showed reduced temporal correlations when returning to the familiar environment (Familiar II) (Fig. 5c). To correct for increased noise deriving from the reduced spatial specificity in the chABC group, we removed all spikes from outside fields and performed the same analysis using only the spikes belonging to grid fields. This did not change the results for neither temporal or spatial pairwise correlations.

**Simulation of grid cell network mimics PNN removal**. Several mechanisms may underlie the reduced inhibitory firing rates

observed after PNN removal, e.g. increased membrane capacitance[31], reduced excitability[30], and increased diffusion of AMPA receptors[29]. We observed structural changes in synaptic input to $PV^+$ cells (Fig. 1d) and reduced inhibitory activity, and we therefore wanted to test how changing synaptic weights in a continuous attractor network would effect the spatial properties of grid cells. We simulated populations of excitatory and inhibitory neurons in a continuous attractor model[47], where excitatory grid cells were connected to distant inhibitory neurons that inhibited excitatory neurons close by (Fig. 6a). Within this network, excitatory neurons only communicate through inhibitory neurons. To generate activity, the excitatory neurons received a non-spatial external excitatory drive from a Poisson spiking generator (Fig. 6b). Because of the competitive inhibitory interaction between the excitatory cells, the activity of the network rapidly settles into a hexagonal grid pattern on the two-dimensional neuronal sheet (Fig. 6c). As expected, reducing excitatory to inhibitory synaptic strength produced a strong reduction in inhibitory firing rates, but it also led to reduced excitatory rates (Fig. 6e), similar to what we observed in the broad-spiking population (Fig. 1g). Grid cells showed increased out-of field rate and thereby lower specificity than control (Fig. 6d). The increased out-of field rate is caused by the reduction in firing rate of the inhibitory units in the simulation (Fig. 6e). Contrary to experimental data, we found increased peak rates of grid fields in the reduced-inhibition model (Fig. 6c). In a separate experiment we created a model where we increased the capacitance of inhibitory units aiming to match findings from an in vitro study where PNNs were removed[31]. Interestingly, in the increased-capacitance model, peak rates were reduced in grid fields similar to what we find in experimental data (Supplementary Fig. 12). All other results from the increased-capacitance model were similar to the results presented in Fig. 6. A combination of effects on inhibitory neurons, i.e. reduced synaptic input and change in intrinsic excitability, is likely to cause the observed effects in grid cell spiking properties in experimental data.

**PNN removal in MEC changes the hippocampal place code**. Grid cells are generally assumed to be the primary determinant of place cell firing[9,15], although this notion has been challenged in both experimental and computational work[11,48,49]. To test if the observed changes in grid cell activity after PNN removal affected the hippocampal place code, we recorded 185 units from hippocampus area CA1 (10 animals of which 4 had been treated with chABC in MEC). In the familiar arena we found that spatial specificity was significantly reduced in animals with disrupted PNNs due to a combination of reduced in-field rate and increased out-of-field rate (Fig. 7b and Supplementary Table 7). These results are similar to what we observed for grid cells in MEC; only changes in in-field and out-of-field rates were more pronounced in place cells.

Next, we introduced the animals to a novel environment to test how place cells responded to decreased stability in MEC. Both control and chABC animals displayed hippocampal remapping. The place cells in the chABC-treated group displayed reduced spatial correlations throughout the entire novel environment recording session (Fig. 7c, d), although individual units were highly variable. Despite reduced spatial correlations, the spatial stability within sessions was almost identical between the groups (Fig. 7e). Notably, we found greatly reduced maximum firing rates in chABC animals in the novel environment compared to controls (median; Control 8.01, $n = 63$; chABC 4.02, $n = 77$, $U = 3916$, $p < 0.001$, Mann–Whitney $U$ test). Maximum firing rates of place cells is normally expected to increase in the novel environment[48,50], which indicates that removing PNNs affected

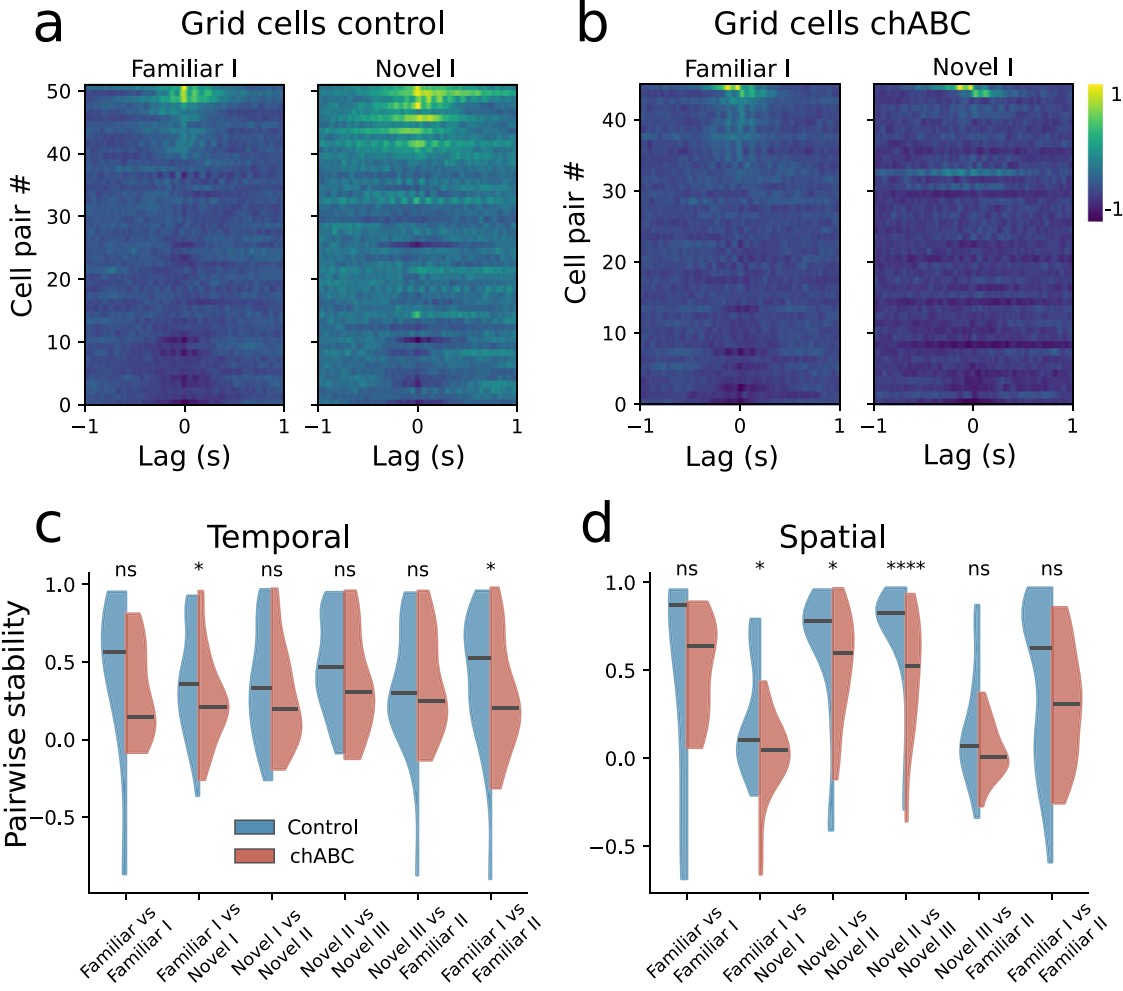

**Fig. 5 PNN removal disrupt the temporal relation between grid cells. a, b** Pairwise temporal cross-correlation of grid cells with brighter color showing stronger correlation. Each line represents a cell pair, sorted by maximum value of the central peak. Values range from −1 to 1. Only cells that could be followed through the novel environment recording sessions were included. **c** Pairwise temporal stability calculated by correlations of pairwise correlations across recording sessions. Grid cells in the familiar environment prior to any change in environmental condition was identified and correlated with the first familiar session at the day of the novel environment experiment. This is seen in the graph as Familiar vs Familiar I. The pairwise stability is significant lower for the chABC-treated group from the familiar to the novel environment (median Familiar I vs Novel I; Control 0.36 ($n = 51$); chABC 0.21 ($n = 45$), $U = 874$, $p = 0.045$). Furthermore, grid cells from chABC-treated rats show reduced correlations in the familiar environment after the animals have explored the novel environment (median Familiar I vs Familiar II; Control 0.52 ($n = 51$); chABC 0.20 ($n = 29$), $U = 529$, $p = 0.036$) (see Supplementary Table 5 for statistics). Violin plots show median (black line); colored shades represent 95th percentile. **d** Pairwise spatial stability calculated across sessions as in **c**. The pairwise spatial correlations are significant lower in chABC-treated rats than controls when going from the familiar to the novel environment (median; Familiar I vs Novel I; Control 0.10 ($n = 51$); chABC 0.05 ($n = 45$), $U = 800$, $p = 0.011$) (Supplementary Table 6). All tests are Mann–Whitney $U$ tests (two sided), *$p < 0.05$, ****$p < 0.0001$. Spatial pairwise correlations are also reduced in the chABC throughout all sessions in the novel environment (Supplementary Fig. 11). Source data are provided as a Source Data file.

the input from MEC sufficiently to impair rate change in hippocampal place cells.

Although PNNs were only removed in MEC, the LFP recorded in hippocampus also showed increased power at the peak of theta (Fig. 7f). Similar to MEC, theta power was increased for all running speeds, but only in the familiar environment (Fig. 7g and Supplementary Fig. 8e). The peak frequency of the theta oscillations increased strongly when the animals' behavior shifted from sessile to movement, but quickly reached a plateau for the control animals, while the injected animals showed a more linear correlation between theta frequency and running speed (Fig. 7h). Interestingly, the correlation between frequency and running speed (frequency score) was decreased in chABC-treated animals in hippocampus, which is opposite from what we see in MEC (Supplementary Fig. 8d).

## Discussion
The grid cell network in MEC has been under intense investigation since its discovery[7], but little is known about how grid cells achieve their extraordinary stability across time and environments. PNNs stabilize synapses and limit plasticity in several brain areas.

We show that enzymatic degradation of PNNs changes grid cell network dynamics by altering the temporal relationship between grid cells and impairing representations of novel environments. This is striking, as the temporal relationship between grid cells are usually very resilient across states and conditions[17,18,51,52]. Our simulations of the grid cell network in a continuous attractor model indicate that reducing the level of inhibition is sufficient to produce impairments in grid cell spatial specificity. Results show that this can arise from reduced

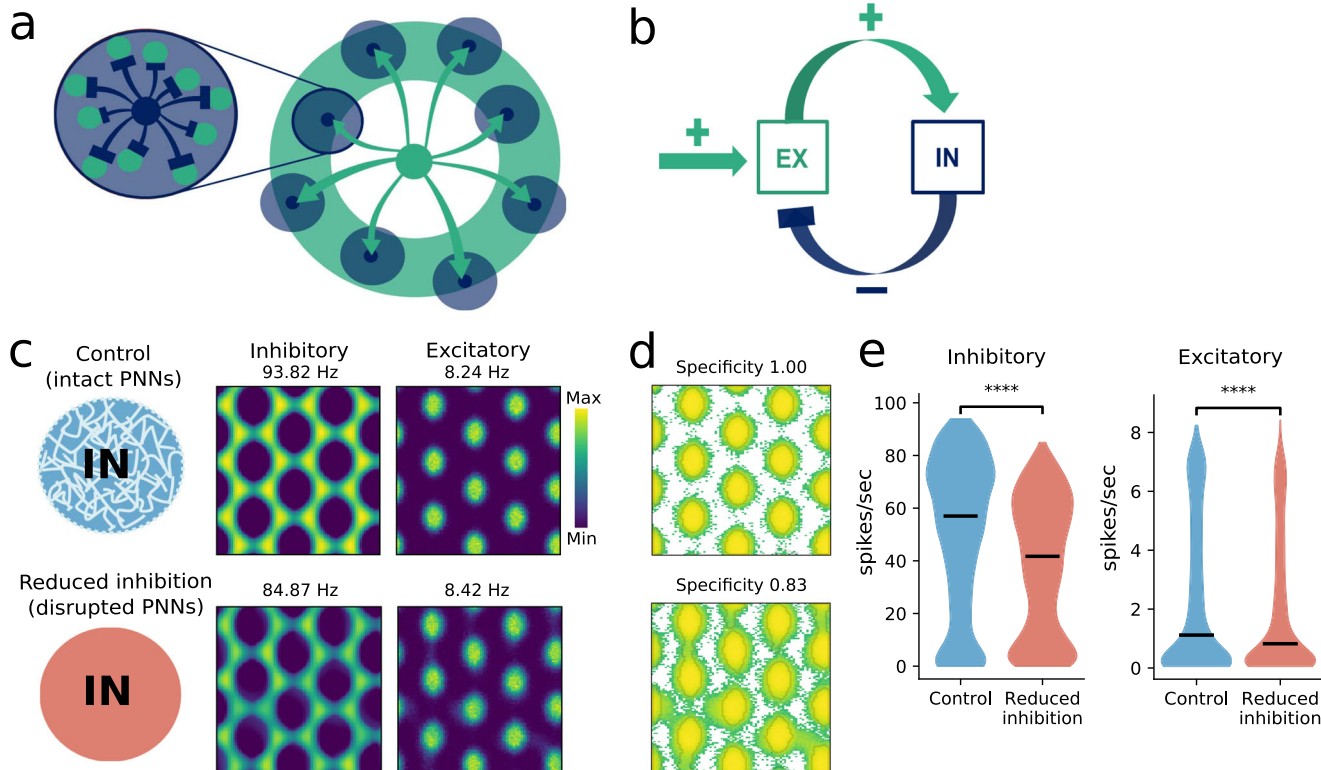

**Fig. 6 Continuous attractor model of the grid cell network with altered inhibition. a** Schematic outline of the model. Each excitatory unit (green) is connected to distant inhibitory neurons (dark blue), which in turn inhibit excitatory neurons locally. **b** A population of excitatory neurons (EX) receive external excitatory drive and feedback inhibition by a population of inhibitory neurons (IN). **c** PNN removal was simulated by reducing the excitatory to inhibitory synaptic strength (lower panel). Rate maps of excitatory neurons show grid pattern for both control and the reduced inhibition network. **d** The grid cells in the reduced inhibition network show increased out-of-field rate and thereby lower specificity than control, as seen with logarithmic color scale. Note the increased number of green spikes representing rate outside the field. **e** The average firing rate of both inhibitory and excitatory neurons are lower in the reduced inhibition network (median, inhibitory control: 57; inhibitory reduced inhibition: 41.7; $U = 29{,}940{,}515$, $n = 6858$ and excitatory control: 1.11; excitatory reduced inhibition: 0.82; $U = 15{,}509{,}839.5$, $n = 5398$, $p < 0.0001$, Mann–Whitney $U$ test (two sided)). Source data are provided as a Source Data file.

excitatory to inhibitory connection strength, similar to what we observe in experimental data after removal of PNNs. These results are in support of the notion that PNNs stabilize the network through its effects on inhibitory, putative PV$^+$ neurons, and suggest a prominent role for PNNs in maintaining the structural and functional organization which shapes the activity of microcircuits in MEC. To target PNNs we used chABC that efficiently degrades the major PNN component, chondroitin sulfate proteoglycans (CSPG). Although chABC is commonly used to probe the function of PNNs, it also affects CSPGs present in the extracellular environment, making it challenging to isolate the effect of PNN removal. Earlier experiments with transgenic mouse knockouts show similar results as chABC treatment[53,54] and patch-clamp experiments after chABC treatment show effects on fast-spiking inhibitory neurons without affecting the intrinsic properties of pyramidal cells in neocortex[55]. Thus, we believe that the effects we report in the current study are due to PNN removal around PV$^+$ cells.

Removal of PNNs in MEC caused synaptic reorganization of punctas onto PV$^+$ cell somas (Fig. 1). When recording from neurons in MEC we found that removal of PNNs caused a reduction in the activity of inhibitory neurons, similar to what has been found in other brain areas[26,54]. This affected the spiking and spatial precision of grid fields in the familiar environment. However, we did not observe impairment in spatial stability or lower gridness scores in familiar environments. This suggests that the connectivity supporting the spatial spiking patterns of grid cells remained stable enough to maintain precise spatial maps as long as the environment was familiar.

Is it possible to separate the effects on network plasticity of PNN removal from the indirect effect it has through the activity level of PV interneurons? Previous work has shown that the PNN is enriched in growth-inhibiting molecules such as Semaphorin 3A[56], and that its removal allows for structural changes[57]. Moreover, increased functional plasticity has been measured after PNN removal in hippocampal CA2 where the PNNs are co-localized with other cell types[58]. At the same time, PV$^+$ neurons are widely thought to be key regulators of network plasticity[22,59]. We believe that our results are a combination of both, where the most obvious plasticity effect from PNN removal is the structural changes in inputs to PV$^+$ cells, but also as network plasticity in novel environment experiments. On a network level, changes likely results from both structural and functional changes to PV$^+$ cells. Previous work show that PNN removal alone does not cause massive alterations to the function, but that this occurs when the network is challenged with new information such as changes in input (e.g. ref. 26). Indeed, in our data, the most pronounced effects of PNN removal were observed when grid representations developed in a novel environment (Fig. 3). Without PNNs, grid cells did not establish stable representations of the novel arena similar to controls. In addition, the novel arena experience interfered with representations of the well-known arena (familiar environment) in animals without PNNs. The latter resembles previous work from the amygdala where extinction training in

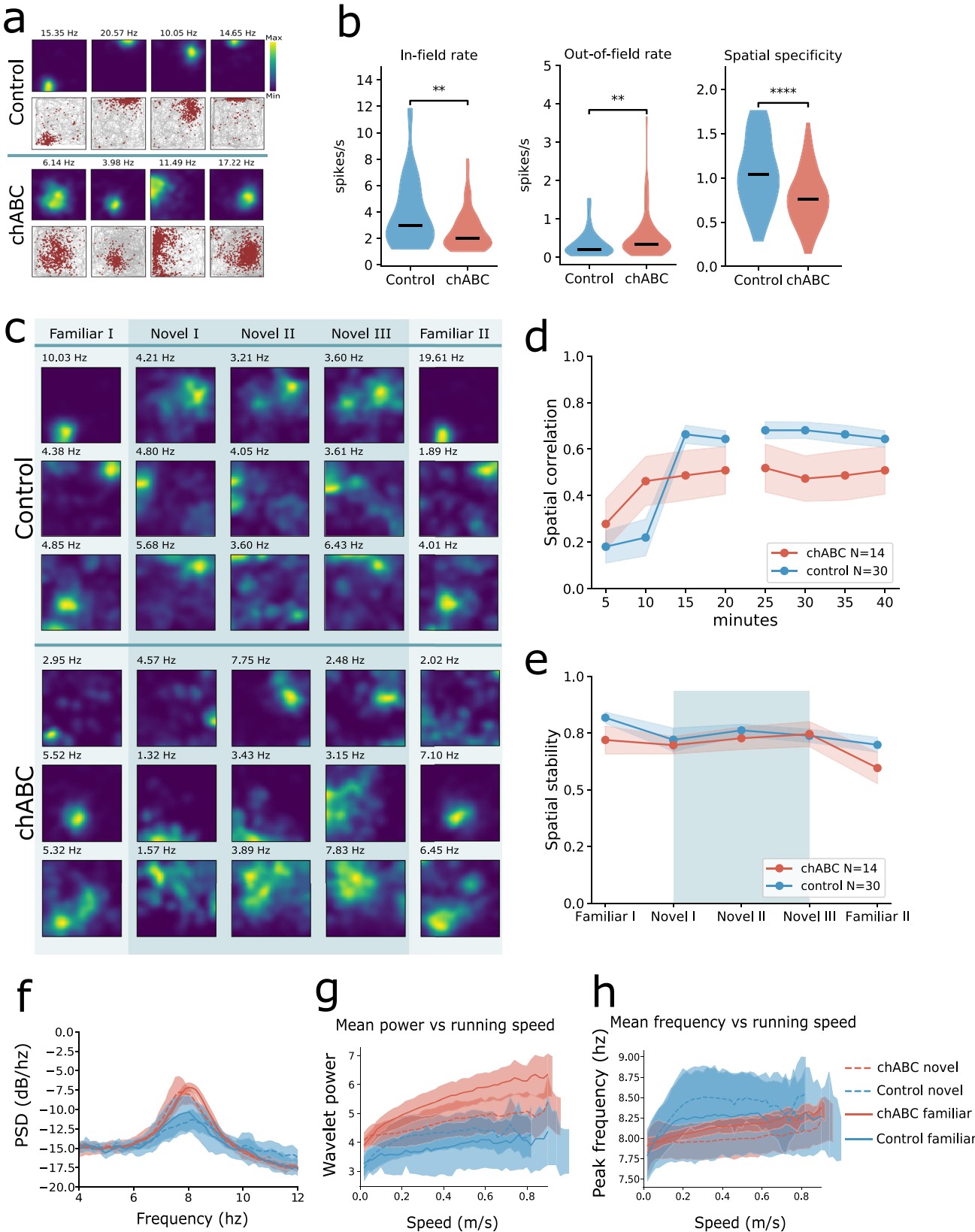

adult animals after fear conditioning, combined with PNN removal, completely replaces the original memory trace instead of creating a separate extinction representation[60]. These data indicate that removing PNNs led to increased plasticity because reduced spatial correlation and reduced gridness were only observed after introduction to the novel environment. Strikingly, grid cells from animals without PNNs continued to show reduced spatial correlation relative to control animals when returning to the familiar environment. However, the same grid maps from the first session in the familiar environment (Familiar I) were

**Fig. 7 Altered hippocampal place code after PNN degradation in MEC. a** Examples of place cells recorded in control and chABC-treated rats. Top row: color-coded rate maps; bottom row: running path with spikes superimposed. Maximum spike rate is denoted above the rate maps. **b** Place cells show reduced in-field rate (median; Control 2.97; chABC 2.03, $p = 0.004$) and increased out-of field rate in chABC-treated rats (median; Control 0.19; chABC 0.34, $p = 0.004$). Violin plots show min to max and median (black line). Width of graph corresponds to number of samples for each value. **$p < 0.01$, ****$p < 0.0001$ (Mann–Whitney $U$ test (two sided)). **c** Rate maps of five consecutive 20 min recording sessions during a novel environment experiment. Rows correspond to individual units and columns to environment condition. **d** Spatial correlation between blocks of 5 min recordings in a novel environment (Novel I and Novel II) against the last 20 min (Novel III) (mean ± s.e.m.; Control 0.65 ± 0.03; chABC 0.46 ± 0.09; main effect of group: $F$ $(1, 40) = 6.011$, $p = 0.0187$, Control $n = 30$; chABC $n = 14$, two-way repeated measures ANOVA with group and time as factors). Points indicate mean; shaded area represents s.e.m. **e** Within a session the place code is stable for both groups measured as spatial correlation between first and last 10 min of the session. **f** The power spectral density (PSD) plot show stronger theta oscillations in chABC-treated animals (red lines) compared to control (blue lines). Lines indicate median; shaded area represent 95% confidence interval. **g** Mean theta power in hippocampus correlated with running speeds from familiar and novel environment recordings. **h** Peak theta frequency increases with running speed. Source data are provided as a Source Data file.

reactivated in the familiar (Familiar II) session after the novel environment exposure albeit less spatially stable. This suggests that the animals kept an intact representation of the familiar environment. It is likely that exposure to a novel environment accelerated plasticity in MEC, possibly leaving synaptic weights prone to modification. This could also have affected stored representations of the familiar environment, thus leading to a temporary alteration of synaptic weights when animals returned to the familiar environment. Together with previously published data, this suggests that intact PNNs facilitate storage and maintenance of separate memory traces so that newly formed memories do not interfere with previously stored experiences. For the grid cell network, this appears as the ability of the network to maintain representations of a seemingly infinite number of different arenas as well as other non-spatial modalities[61].

In contrast to the highly organized representations in MEC, different sub-populations of place cells in the hippocampus are recruited to represent different environments[62,63]. It is likely that the connectivity and high levels of plasticity in the hippocampal network enables place cells to rapidly reorganize and produce a vast number of orthogonal representations. While place cells can form even if input from MEC is minimized, they display reduced spatial precision and stability under such circumstances[9]. Specifically increasing the scale of grid cells or impairing grid cell spiking patterns reduce the long-term stability of place cells[48,64], making it evident that the properties of grid cell input play an important role for place cell spatial coding. In support of this, our data from the hippocampus also show that removing PNNs in MEC leads to changes in recorded place cells.

When the chABC-treated rats were introduced to the novel environment, we observed reduced spatial correlations in the place cell population similar to our observations for grid cells. This suggests that the activity pattern of grid cells influences the place cell coding of novel environments under normal conditions. Taken together, our data from hippocampus when PNNs were removed in MEC suggest that place cells rely on adequate grid cell activity for precise spatial representations in both familiar and novel environments.

We show that theta power is increased in the chABC-treated animals. It is likely that the decrease in burst events and spiking variability give rise to more synchronous firing patterns leading to the increased theta power observed in animals treated with chABC. Since increased theta power is associated with enhanced plasticity and increased synchronization of interconnected input[65,66], we find it likely that the high theta power in combination with reduced inhibitory activity represents a state of increased plasticity in the MEC network.

Interestingly, the increase in theta power was also observed in the recordings from CA1 of the hippocampus. Both hippocampus and entorhinal cortex receive theta rhythmic drive from the medial septum. However, our findings suggest that in addition to input from medial septum, hippocampal theta is strongly influenced by the theta in MEC.

It has been suggested that environmental novelty is signaled by a sharp reduction of the hippocampal theta frequency, which gradually disappears with repeated exposures to the environment[67]. In line with this we observed a reduction in frequency between the familiar environment (Familiar I) and the novel environment (Novel I), for both control and chABC-treated animals (Fig. 4). As expected, the subsequent recordings in the novel environment (Novel II and III) the peak theta frequency increased as the environment became more familiar although the increase was faster and more pronounced in the control animals. Furthermore, the control animals returned to their baseline frequency when returning to the familiar environment (Familiar II), while the theta frequency continued to increase in most chABC-treated animals.

We found that spatiotemporal correlations between pairs of grid cells are maintained in control animals when they are introduced to a novel environment, but not in animals where the PNNs were removed (Fig. 5). To our knowledge, this is the first manipulation to alter the spatiotemporal relationship between grid cells across environments. It is well established that the relationship between grid cell pairs remains highly stable across environments despite large changes in single-cell responses[17,18]. Furthermore, grid cells exhibit a temporal correlation structure that transcends across behavior states[51,52]. This indicates that the local circuitry supporting grid cell firing normally exhibits little plasticity, possibly due to the presence of PNNs.

Strikingly, we also find that pairwise spatial correlations are reduced in chABC-treated animals for all sessions in the novel environment (Supplementary Fig. 11). This provides further support for the observations of reduced stability of newly formed grid fields after PNN removal. The reduced pairwise correlations could both be a result of lower inhibition and changes in synaptic connections onto inhibitory neurons. Either possibility is likely to result in an increase in network plasticity.

Continuous attractor models have shown that inhibitory connections are sufficient to generate the hexagonal structure of the grid pattern[19,37,68]. The dispersed grid cell patterns we observed after PNN removal might therefore be a consequence of reduced inhibition. In particular, this is likely since the increased out-of-field spiking occurs where grid cells presumably rely on inhibitory domination under normal circumstances. This result is consistent with a previous report showing that pharmacogenetic silencing of PV$^+$ interneurons selectively affects grid cells in a similar manner[14]. However, in contrast to our findings, this study observed an increase in mean spike rate of grid cells, but no change to maximum spike rate. This discrepancy suggests that removal of PNNs causes alterations in network properties that cannot be explained solely by an acute reduction in PV$^+$ cell activity. In addition, we found reduced spatial specificity and maximum

firing rate (both spatial and temporal). Together this means that the effects of PNN removal is not merely an increase in unspecific firing outside fields (as reported by Miao et al., 2017 when acutely reducing PV$^+$ cell activity), but also a reduction in the activity within grid fields. This contrasts the findings from Miao et al. (2017)[14], where inside field activity was unchanged. If stable attractor states would rely only on PV$^+$ cell activity one should observe reduced spatial stability also in the familiar environment prior to novel environment exposure. Miao and colleagues found reduced spatial stability in grid cells as long as PV$^+$ cells were inhibited. However, their reduction of PV$^+$ cell activity is much larger compared to the slight reduction we observed after PNN removal. We did not find instability of representations in familiar environments, and because the change in stability occurs only during and right after the network encodes new information, we find it unlikely that changes in activity levels of PV$^+$ cells are sufficient to explain our results from novel environment experiments.

In line with experimental data, we observed that lowering excitatory (E) synaptic strength onto inhibitory (I) neurons caused reduced inhibitory spiking in the continuous attractor model (Fig. 6c). Furthermore, it also led to a decrease in mean spike rate of excitatory neurons, similar to what we observed in the experimental data. This result is, at first glance, counter intuitive as reduced inhibition would be expected to increase excitatory activity. However, the dynamical interactions in the model and the spatial outreach of connections have a large influence on activity levels of both inhibitory and excitatory neuron populations.

We also performed another simulation where we increased the capacitance of inhibitory neurons in line with Tewari et al. (2018)[31]. These results are similar to the simulation with decreased EI weights except in maximum rate, while the change in capacitance leads to an increase, the change in EI weights leads to a decrease. In this sense the capacitance model is more in line with experiments; however, the change in experimental data is likely to be an effect from the change in bursting. Due to the choice of model neuron (exponential integrate and fire) the model does not support bursting, and is thus not necessarily comparable to experimental results regarding maximum rate. Rather, results from these two models indicate that change in inhibitory firing rate from experimental familiar data may be a combined effect from change in capacitance and EI weights.

Our results from the familiar environment experiments could largely be explained by reduced firing rates in both inhibitory and excitatory neurons, as shown by the in silico data. However, in a novel environment, increased plasticity seems to be relevant because reduced spatial correlation and reduced gridness are not observed until the animals are introduced to a novel environment. In addition, disrupted stability in cross-correlation structure is clearly seen when there is a large change in the environment, i.e. when the animal is moved from Familiar I to Novel I and later when reintroduced to the familiar environment.

Accumulating evidence points to a low-dimensional continuous attractor to be present in the grid cell network[18,51]. In this theory, the co-activity of pairs of neurons remains stable throughout experiments of varying environmental conditions, meaning that the population activity resides in a low-dimensional manifold. Our results showing destabilization of pairwise correlations indicate that this manifold is partially disrupted, suggesting that the attractive dynamics of the population become less pronounced. It is not clear, however, why this only occurs when the animal is introduced to a novel arena. One possibility could be that exploration in a novel environment stimulates synaptic plasticity which normally is restricted in adult animals.

Other computational models have shown that the spatial firing pattern of grid cells can develop and be supported by external input from place cells[49,69]. However, in order for grid cells to have a shared alignment as shown in numerous experimental studies[17,18,51,52], it seems likely that some form of attractor dynamics is necessary[70]. The seemingly hardwired nature of the MEC L2/3 network could therefore be necessary to maintain the relative relationship between grid cells.

By removing PNNs, we show that the spatial representations of individual grid cells remain relatively stable, while pairwise spatiotemporal relationships decrease in strength. Hence, these data provide support for a local, low-plasticity attractor network and the role of inhibitory control in grid cell firing postulated by continuous attractor models[19,37,68].

In summary, our data show that PNNs are essential for stabilizing the grid cell network and to ensure proper function when the network is challenged to create representations of novel environments. By stabilizing connections and supporting proper inhibition, intact PNNs may be essential to support efficient navigation.

## Methods

**Subjects**. This study used recording data from 23 male Long-Evans rats (3–8 months old, 350–550 g at surgery). After surgeries the animals were housed individually in transparent Plexiglas cages ($45 \times 30 \times 35$ cm) in a temperature- and humidity-controlled vivarium. All rats were maintained on a 12-h light/12-h dark schedule. Testing occurred in the dark phase. The rats were kept at 85–90% of free-feeding body weight and food deprived 18–24 h before each training and recording trial. Water was available ad libitum. Experiments were performed in accordance with the Norwegian Animal Welfare Act and the European Convention for the Protection of Vertebrate Animals used for Experimental and Other Scientific Purposes.

**Surgical procedures**. All surgical procedures were performed in an aseptic environment. Rats were anesthetized with isoflurane mixed with air (5% induction, 1.5–2% for maintenance) and immobilized in a stereotaxic frame (World Precision Instruments Ltd, Hertfordshire, UK). They were given subcutaneous injections of buprenorphine (0.04 mg/kg) and local subcutaneous injections of bupivacaine/adrenaline (Marcain adrenaline, 13.2 mg/kg) in the scalp before surgery began. The scalp was shaved and cleaned with ethanol and chlorhexidine. Heart rate and core temperature were continuously monitored throughout the operation through a MouseStat system (Kent Scientific, CT, USA), the latter in a feedback mechanism to a heating pad. In addition, the hind paw withdrawal reflex was used to assess the depth of anesthesia.

Craniotomies were made bilaterally above the MEC, using a hand-held Perfecta-300 dental drill (W & H Nordic, Täby, Sweden). Injections were made in concomitance with tetrode implantation. Protease-free chondroitinase ABC (chABC) from *Proteus vulgaris* was purchased from Amsbio (Abingdon, UK) and diluted in filtered 1× phosphate-buffered saline (PBS) to a concentration of 0.05 U/µl[25]. Glass pipettes with a 15–20-µm opening diameter were backfilled with mineral oil, assembled into a NanoJect II microinjector (Drummond Scientific Company, PA, USA), and loaded with chABC. Injections were done in two depths at each location (2500 and 3200 µm below the dura). The locations were at AP 0.5 anterior of the transverse sinus, and ML 4.4 and 4.7 mm relative to the midline. A total volume of 3.1 µl was injected in each hemisphere. The injections were made step-wise over 2 min at each position, and the pipette left for another 2 min before retraction.

Tetrodes were implanted above MEC at AP 0.4 ± 0.1 mm in front of the transverse sinus and ML 4.5 ± 0.1 mm relative to the midline. Tetrodes implanted above hippocampus were placed at AP Bregma −3.8 ± 0.2 mm and ML 3.1 ± 0.1 mm. The depth of implantation was 1800 µm measured from the surface of dura. Jeweler's screws fixed to the skull served as ground electrodes. The microdrives were secured to the skull using jeweler's screws and dental cement. All animals were given a subcutaneous injection of carprofen (5 mg/kg) at the end of the surgery, and the edge of the wound was cleaned and local anesthetic ointment Lidocain was applied. This was repeated for 3 days after surgery.

**Extracellular recordings**. Electrophysiological recordings were performed within 14 days after chABC injections since it has been shown that less than 40% of PNNs are reassembled by that time[26]. The recording system used was daqUSB, provided by Axona (Herts, UK). Signals were amplified 8000–15,000 times and band-pass filtered between 0.8 and 6.7 kHz. Triggered spikes were stored to disk at 48 kHz (50 samples per waveform, 8 bits/sample) with a 32-bit timestamp (clock rate at 96 kHz). Spike waveforms above a threshold of 50 µV were time-stamped and digitized at 32 kHz for 1 ms, and saved to the hard-drive for offline analysis. One

channel in each hemisphere served as a reference electrode to record LFP; LFP signals were amplified 3000 times, low-pass filtered at 500 Hz, and stored at 4.8 kHz (16 bits/sample).

Prior to surgery the animals were habituated to a recording environment that during the 14 days of experiments became their familiar environment. The environment consisted of a $1 \times 1$ m black box containing a white A4 sheet on one of the walls serving as a local cue. Animals were motivated to explore the environment by chocolate crumbles that were thrown at random intervals into the recording arena. Before starting novel environment recordings, we conducted daily recordings in the familiar environment [usually two or three per day per animal], where we searched for grid cells. During these recording sessions, animals ran in the arena for sessions lasting from 10 to 20 min (depending on how fast they could cover the entire environment). Between sessions, the tetrodes were adjusted downwards in steps of 50 μm and the animals allowed to rest in their home cage for 20–30 min. If only a few grid cells were recorded or the animal did not show a strong motivation for exploring the environment, familiar environment recordings were continued (within 14 days) until we considered both the number of grid cells recorded simultaneously and the animal behavior to be satisfactory for initiating novel environment experiments.

When the tetrodes were located in MEC (usually when three or more grid cells could be recorded simultaneously), novel environment experiments were conducted. The novel environment consisted of an identical recording setup as the one in the familiar environment but located in a different experiment room. In addition, the cue card was placed on a different wall in the recording arena relative to the familiar environment. This paradigm is shown to cause consistent global remapping of spatial representations in both hippocampus and MEC[17,62].

First, a recording session of 20 min was conducted in the familiar environment followed by $3 \times 20$ min recording sessions in a novel environment. Lastly, the animals were put back in the familiar environment for an additional 20 min session. The animals rested for 5–10 min in the home cage between sessions. This experimental procedure was repeated for three consecutive days, with the exception that the novel environment recording on days 2 and 3 consisted of only one 20 min long recording session.

Single units were isolated using the offline manual spike sorting software Tint (Axona). Clusters were manually isolated using 2D projections the multidimensional parameter space (waveform amplitudes). Cross-correlation and autocorrelation were used as additional tools for assessing the quality of separation. Spikes falling within the 2–3 ms refractory period of neurons were considered to belong to other units and if further spike sorting did not remove these, the unit was discarded. Identification of the same units in successive recording sessions (novel environment experiments) was done by manual inspection of cluster location and waveform, in addition to comparing the behavioral outputs of single units.

**Tissue processing and immunohistochemistry.** At the end of the experiments, animals were deeply anesthetized by an intraperitoneal injection of pentobarbital sodium (50 mg/kg) and intracardially perfused with 0.9% NaCl followed by 4% paraformaldehyde (PFA) in $1 \times$ PBS. The brains were dissected out and post-fixed in 4% PFA overnight. They were cryoprotected in 30% sucrose in $1 \times$ PBS for 3 days, and 40 μm sagittal (MEC) or coronal (hippocampus) sections were cut with a cryostat. Staining procedures were performed on free-floating sections under constant agitation unless mentioned otherwise. The lectin WFA was used to visualize PNNs.

The sections were rinsed times in $1 \times$ PBS and blocked with 2% goat serum with 0.3% Triton X-100 in $1 \times$ PBS for 1 h at room temperature. The sections were then incubated with biotin-conjugated WFA (#L1516, Sigma-Aldrich Chemie, 1:200) in blocking solution overnight at 4 °C. On the following day, the sections were rinsed with 0.3% Triton X-100 in $1 \times$ PBS, endogenous peroxidase activity was quenched for 3 min with 2% $H_2O_2$ in $dH_2O$, and the sections incubated with ABC solution (ABC Peroxidase Standard Staining Kit, Thermo Fisher Scientific). After rinsing the sections in Tris nonsaline (TNS), staining was visualized by adding a 3,3-diamonobenzidine hydrochloride (Sigma-Aldrich Chemie) solution (0.67 mg/ml 0.05 M Tris-HCl, with 0.8 μl $H_2O_2$/ml) for 3–10 min. Sections were rinsed in TNS, mounted on coverslips, and dried for 1 h. They were then left overnight in a solution of 1:1 choloroform and 96% ethanol. The following day, the sections were rinsed in $dH_2O$ and counterstained using Cresyl Violet and Nissl substance using Cresyl Violet and coverslipped with Entellan. Tetrode tracks were identified, measured, and photographed through an Axioplan 2 microscope (Carl Zeiss, Oberkochen, Germany). High-resolution images were then stitched together using the MosaiX extension in the AxioVision software (Carl Zeiss, Oberkochen, Germany).

A subset of sections from each animal were used for visualizing the chABC-treated area in MEC. Sections were prepared as described above. After blocking, the sections were incubated with WFA (#L1516, Sigma-Aldrich, 1:200) and chondroitin-6-sulfated stubs (MAB 2035, Millipore, 1:1000) in blocking solution overnight at 4 °C. The next day the sections were rinsed with 0.3% Triton X-100 in $1 \times$ PBS and incubated for 2 h with Streptavidin Alexa 488 (#S-11223, Life, 1:1000) and donkey anti mouse Alexa 594 (#A-21203, Life, 1:1000) in $1 \times$ PBS. Sections were rinsed in $1 \times$ PBS, mounted to Superfrost Plus slides, washed in $dH_2O$, and coverslipped with FluorSave Reagent (#345789, Millipore).

Animals were injected with chABC in MEC of one hemisphere and artificial cerebrospinal fluid (aCSF) in one hemisphere. After surgery, animals were kept in their home cage for 5 days until they were transcardially perfused and the brains were sectioned as described above. Prior to antibody incubation, sections were blocked with 2% BSA and 0.3% Triton X-100 in $1 \times$ PBS for 1 h.

Two sections from each hemisphere representing the medial-center and lateral-center parts of the injection site (approximately 100–120 μm apart) were incubated overnight with goat anti-PV (#PVG-214, Swant, 1:2000), biotin-conjugated *Wisteria floribunda* agglutinin (#L1516, Sigma-Aldrich, 1:200), and either rabbit anti-vGlut1, rabbit anti-vGlut2 or rabbit anti-vGat (1:5000, synaptic vesicle antibodies kindly donated by Dr. Farrukh Chaudry, UiO) in blocking solution containing 0.2% sodium azide. The next day, sections were rinsed with $1 \times$ PBS and incubated with donkey anti-goat Alexa 594 (#A-11058, Life, 1:1000), Streptavidin Alexa 647 (#S-21374, Life, 1:1000), and chicken anti-rabbit Alexa 488 (#A-21441, Life, 1:4000) in $1 \times$ PBS for 2 h. Sections were rinsed in $1 \times$ PBS, mounted to Superfrost Plus slides, washed in $dH_2O$, and coverslipped with Fluoromount mounting medium containing DAPI (#0100-20, AH Diagnostics).

Images were acquired with an Andor Dragonfly spinning disc microscope using the Fusion software, with a Zyla 5.5 sCMOS camera covering $2048 \times 2048$ pixels. Overview images of entire slides containing two sections per antibody combination from one hemisphere of one animal were acquired using a 4× objective (NA 0.2). This verified that chABC had degraded all PNNs in mEC in all animals, and verified the needle tracks from aCSF injections in sham-treated tissue.

Landmarks from these images were used to verify that all high-magnification images were acquired from within the mEC. Detailed images of $PV^+$ somas and markers of synaptic vesicular transporters were acquired through a ×60 objective (1.4 NA). Z-stacks with a step size of 0.16 μm were acquired through the entire cell soma. One to three somas were included in each stack.

Images were analyzed in a similar fashion to ref. [71] using Imaris 9.1.2. First, regions of interest containing each soma were selected. Second, synaptic boutons were detected using the built-in spot detection algorithm with background subtraction. Finally, boutons in contact with each respective soma were labeled using the Spots close to Surface tool, with maximum distance from center of the spot to the surface set at 0.8 μm. Each set of spots and soma were manually inspected and revised after processing. Data from all animals in each group (chABC or aCSF) were pooled prior to statistical analysis.

**Statistics.** All tests are two sided. When testing differences in means between the experimental groups, we tested for normality in a subset of data using the Shapiro–Wilk test in Sigmaplot. For the subset of data we tested, normality failed in all, hence we only used non-parametric tests (Mann–Whitney $U$ test) for all data. To test if grid cell parameters were consistent across animals within a group and that results were not caused by random variations, we performed a permutation resampling test in addition to Mann–Whitney $U$ test. A low $p$ value allowed us to reject the hypothesis that data could be randomly drawn from either group. For testing spatial correlations over time in the novel environment, we performed ANOVA with repeated measures and Šidák's multiple comparisons post hoc test, using GraphPad Prism 8. All other analyses were performed in Python.

**Rate maps.** To produce spiking rate maps, we divided the arena into equally sized bins of 2 cm × 2 cm. For each bin, we counted the number of spikes and the time spent within the bin to produce a spike map and an occupancy map, respectively. We further smoothed the spike map and the occupancy map individually by using the convolution of a two-dimensional Gaussian kernel. By dividing the value of each bin in the smoothed spike map by the corresponding bin in the smoothed occupancy map, we produced a smoothed rate map in two versions, one with standard deviation $\sigma = 3$ cm and one with standard deviation $\sigma = 5$ cm.

To create a spatial autocorrelogram used for visualization and subsequent analyses, we correlated the smoothed rate map with itself using the `scipy.signal.fftconvolve` function from the SciPy package with the mode parameter set to mode = "full".

**Gridness score.** To calculate the gridness score of each rate map, we first identified the central peak and the six closest peaks in the autocorrelogram by finding the maxima of the autocorrelogram and calculating the distance to the center of the autocorrelogram to each maximum. The center peak was selected as the peak closest to the center of the autocorrelogram. We then masked out the center peak with a disk centered on the center peak with a radius of half the distance to the closest peak. To mask the area of the autocorrelogram outside a circle going around the outside of the outermost of the six closest peaks, we masked everything outside a disk centered on the center peak with a radius 3/2 times the distance to the outermost of the six closest peaks. Next, the masked autocorrelogram was rotated by increments of 30° up to 150°. For each rotation, we calculated the Pearson product moment correlation coefficient against the originally masked auto-correlogram. Finally, the gridness score was calculated by taking the lowest coefficient found with rotations 60° and 120° and subtracting the highest coefficient found with rotations 30°, 90°, and 150°.

To obtain null distributions such that statistical properties, such as gridness, could be measured against, we generated a randomized spike train for each registration using the function `spike_train_surrogates.dither_spike_train` from the Elephant electrophysiology analysis toolkit (neuralensemble.org/elephant). We generated $n = 1000$ randomized spike trains for each session using a shift of 30 s and with the `edges` keyword set to `edges=True`.

**Spatial correlations**. Spatial correlation across trials was calculated according to the procedure for calculating inter-trial stability in ref. [39]. To calculate the reliability of spatial firing between entire or partial trial against a reference, we calculated the Pearson product moment correlation coefficient of the rate map from the trial against the rate map of the reference using the `corrcoef` function from the NumPy Python package.

Spatial stability was calculated as within-trial spatial correlation according to the procedure for calculating intra-trial stability in ref. [39]. To calculate the reliability of spatial firing within a trial, we calculated the Pearson product moment correlation (see inter-trial stability above) of the first 10 min of the trial against the last 10 min of the trial.

**Spatiotemporal pairwise cross-correlations**. Grid cell pairs were selected from the same recording and hemisphere. Temporal pairwise cross-correlations were calculated by first calculate the instantaneous firing rate by convolving the spike times with a Gaussian kernel 10 ms width using `statistics.instantaneous_rate` (http://neuralensemble.org/elephant/) at temporal differences from −1 to 1 s using `numpy.correlate` with `mode="full"` and `method="fft"`. The rates were z-scored and one of the two z-scored rates were scaled by its size to achieve a cross-correlation value between −1 and 1. Then, the Pearson correlation coefficient of pairwise cross-correlations was used to quantify the similarity across experimental states using `numpy.corrcoef`.

To assess the spatial cross-correlations we first calculated the rate maps, then for each pair we calculated the 2D cross-correlation function using `astropy.fft_convolve` (https://www.astropy.org/) with `boundary="wrap"`, `normalize_kernel=False, nan_treatment="fill"`. Each rate map were z-scored and one of the two z-scored rates were scaled by its size to acheive a cross-correlation value between −1 and 1. Then, the Pearson correlation coefficient of pairwise cross-correlations (reshaped to 1D) was used to quantify the similarity across experimental states using `numpy.corrcoef`.

To control for possible effects due to changed theta activity in the experimental group a Butterworth bandstop filter was used to filter out frequencies between 4 and 10 Hz, this did not affect results significantly. Furthermore, to control for effects due to differences in out of field spikes between the experimental and control groups we also removed all spikes landing outside fields before calculating rates, this did also not change results significantly.

**Identification of firing fields**. To estimate the firing within and outside fields, we first identified the individual fields in the rate map. Following the protocol of ref. [72], we first identified a global field radius as 0.7 times half the distance from the center peak to the closest peak in the autocorrelogram. Next, we identified all the peaks in the rate map before excluding the lowest of any two peaks within a distance shorter than the global field radius.

To define the extent of each field, we calculated the Laplacian, $\nabla^2$, of the smoothed rate map to obtain its curvature and excluded regions with a positive Laplacian, which are the valleys of the rate map. The Laplacian was calculated using the `ndimage.laplace` function from the SciPy Python package.

To separate and label the remaining regions, we used the `ndimage.label` function. Any regions with an area less than 9 bins were excluded. The areas were then sorted based on the mean firing rate in each area.

Fields were defined to be any labeled region found after taking the Laplacian that corresponded with a non-excluded peak from the protocol of ref. [72]. The fields were used in subsequent analyses to identify in- and out-field spikes.

Spatial specificity is a measure of the firing ratio between in-field firing and out-field firing and is defined as

$$\text{spatial specificity} = \log_{10}\left(\frac{\text{FR}_{\text{in−field}}}{\text{FR}_{\text{out−field}}}\right). \tag{1}$$

Spatial information is an estimate of to which degree the animal's position can be predicted based on the firing of the cell and is given in bits per second. Spatial information was found by

$$\text{spatial information} = \sum_i p_i \frac{\lambda_i}{\lambda} \log_2 \frac{\lambda_i}{\lambda}, \tag{2}$$

where $p_i$ is the probability of the animal being in bin $i$, given by the occupancy map divided by the total session time, $\lambda_i$ is the firing rate in bin $i$, and $\lambda$ is the mean firing rate.

**Coefficient of variation**. The CV for ISI was calculated as the standard deviation of the ISI distribution $\sigma_{\text{ISI}}$ divided by the mean of the ISI distribution $\mu_{\text{ISI}}$:

$$\text{CV} = \frac{\sigma_{\text{ISI}}}{\mu_{\text{ISI}}}. \tag{3}$$

To reduce bias in ISI that arises from animals traveling between receptive fields, we calculated only in-field CV-values by extracting one CV value for each pass through the fields. This was compared with two other methods. In the first, a CV value was calculated for the entire session. The second followed the protocol of ref. [68], where sections of the recording where the speed of the rat was less than 8 cm/s were selected. A CV value was then calculated for the sections with a longer duration than 0.2 s and at least two spikes. See Supplementary Fig. 5 for comparison of the three methods.

**Bursting**. The bursting ratio was estimated by taking the ISI and defining any spike with a pre-ISI of more than 10 ms and a post-ISI <10 ms as the start of a bursting event. Any spike with a pre-ISI and post-ISI of more than 10 ms was identified as a single-spike event. All other spikes were assumed to be part of a bursting event and were not counted. The bursting ratio is then given by the number of bursting events $n_{\text{B}}$ divided by the number of single-spike events $n_{\text{S}}$:

$$\text{bursting ratio} = \frac{n_{\text{B}}}{n_{\text{S}}}. \tag{4}$$

**Spike waveform**. To separate between narrow- and broad-spiking units we calculated the time from through to peak and time to cross the half-width amplitude of the largest amplitude through from the mean waveform. In peak to through calculations, sampling period of each spike was increased 200-fold by cubic interpolation to get an accurate measure of peak times. The half-width crossing time was refined by a linear interpolation between crossings of the constant line of half amplitude. All interpolations were done with `scipy.interpolate.interp1d`.

Finally we separated clusters with `scipy.cluster.vq.kmeans` separately on the control and chABC group.

**Speed score**. The speed score was computed according to Kropff et al.[4], briefly described below. To compute the correlation between running speed and the firing rate of neurons we first calculated the instantaneous speed and interpolated linearly to match the sampling frequency of the firing rate. The firing rate was computed by convolving a Gaussian kernel with the spike times using `statistics.instantaneous_rate` and `kernels.GaussianKernel` from the Elephant (github.com/NeuralEnsemble/elephant) package. Finally the speed score was computed as the Pearson correlation coefficient between the instantaneous speed and the firing rate for speed between 0.02 and 1.0 m/s.

**Head direction**. The head direction was computed from the relative angle between two LEDs attached to the microdrive. Then the instantaneous firing rate from a neuron was computed as a function of angle and used to compute the average firing rate as a function of angle using `numpy.histogram`. Finally the head direction score was computed with `pycircstat.resultant_vector_length` using the PyCircStat(github.com/circstat/pycircstat) package.

**LFP spectrum analysis**. To calculate the power spectrum versus running speed ($v$) we first calculated the instantaneous speed and interpolated linearly to match the sampling frequency of the LFP signal at 250 Hz. The power spectrum was calculated by means of a continuous wavelet transform on the z-scored LFP signal using a Morlet wavelet with nondimensional frequency $\omega_0 = 80$ (ref. [73]) and the PyCWT library (https://github.com/regeirk/pycwt). Then a weighted histogram of of speed $v$ with bin size 0.02 m/s was calculated in the range $v \in [0.02, 1]$ m/s. The weights were either the mean power of the wavelet spectrogram over frequencies $f \in [4, 12]$ or the frequency at the maximum power within the same frequency range.

The PSD were calculated on z-scored LFP signal using the Welch method given by `mlab.psd` from the Matplotlib package[74].

The confidence intervals presented in Fig. 4a–c were calculated by bootstrapping each data point from each recording at the 95% confidence level.

**Continuous attractor model**. Simulations were conducted to assess the effects of reduced inhibition either due to a reduction in excitatory to inhibitory connections or increased membrane capacitance, as reported by Tewari et al.[31], due to removal of PNNs. To this end populations of excitatory and inhibitory neurons were simulated in a continuous attractor model. Neurons were modeled by the exponential integrate and fire model given as

$$C_{\text{m}}\frac{\text{d}V}{\text{d}t} = -g_{\text{L}}(V - E_{\text{L}}) + g_{\text{L}}\Delta_{\text{T}}\exp\left(\frac{V - V_{\text{T}}}{\Delta_{\text{T}}}\right) + I_{\text{syn}}. \tag{5}$$

When the membrane potential $V$ reaches the threshold $V_{\text{T}}$ there is an exponential increase in potential, with slope factor $\Delta_{\text{T}}$. The nonlinear spike is reset at $V_{\text{reset}}$

whereas the potential is reset to the equilibrium potential $E_L$. Moreover, $C_m$ denotes the membrane capacitance, $g_L$ represents the membrane leak conductance.

The synaptic input current $I_{syn}$ is modeled by a difference of exponentials, also known as the $\beta$ function given by

$$I_{syn}(V, t) = \sum_i g_i(t)(V - E_{rev,i}). \tag{6}$$

Here, $E_{rev}$ is the reversal potential, $i$ denotes neuron number, and the conductance $g$ is given by

$$g_i(t) = w_i(r)K(e^{-(t-t_0)/\tau_{decay}} - e^{-(t-t_0)/\tau_{rise}}). \tag{7}$$

In Eq. (7), $w(r)$ is a spatial connectivity function, $t_0$ represents onset time of a incoming spike, and $\tau_{decay}, \tau_{rise}$ are the decay- and rise time-constants, respectively. Finally, $K$ is a normalization factor, described in detail in ref. [75] [Eq. 6.6].

To reach a continuous attractor state with hexagonal bump pattern, excitatory and inhibitory neurons were placed in grids of $100 \times 100$ each population forming a layer of extent $0 < x < 2\pi$, $0 < y < 2\pi$. Excitatory neurons were driven by a background Poisson noise process $p_{rate}$ and connected to the inhibitory layer through a doughnut shaped connectivity pattern with spatial connectivity function given by

$$w_{EI}(r) = \begin{cases} \bar{g}_{EI}, & r_{EI,inner} < r < r_{EI,outer}, \text{ for } 0 \le \theta < 2\pi \\ 0 & \text{elsewhere} \end{cases} \tag{8}$$

Here $\bar{g}$ is the peak conductance amplitude, subscript EI indicates that the connectivity is directed from excitatory to inhibitory population. Position is given in polar coordinates with radial length $r = \sqrt{(x - x_0)^2 + (y - y_0)^2}$ relative to neuron position $x_0, y_0$, and angle $\theta$. Inhibitory neurons were connected to excitatory neurons by

$$w_{IE}(r) = \begin{cases} \bar{g}_{IE}, & 0 < r < r_{IE,outer}, \text{ for } 0 \le \theta < 2\pi \\ 0 & \text{elsewhere.} \end{cases} \tag{9}$$

Topological connections were specified using the Topology package in NEST, and simulations were performed with NEST[76]. For parameters used in the model, see Supplementary Table 11.

**Reporting summary**. Further information on research design is available in the Nature Research Reporting Summary linked to this article.

## Data availability
The raw data are available from the corresponding author upon reasonable request. Source data are provided with this paper.

## Code availability
The source code used for analysis are available from the corresponding author upon reasonable request.

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

## Acknowledgements

This work was funded by the Research Council of Norway (Grant No. 231248 to T.H. and grant No. 217920, 248828, 250259 to M.F) and the University of Oslo's Strategic Research Initiative CINPLA. The authors wish to thank Anne Marthe S. Kvello for help with pilot experiments, and Jennifer Hazen and Benjamin A. Dunn for discussions and useful suggestions to the manuscript.

## Author contributions

A.C.C., M.F., and T.H. conceived and designed the project; A.C.C., K.K.L., and J.S.B. conducted the experiments; M.E.L. performed the simulations; A.C.C., K.K.L., M.E.L., S.-A.D., H.S., and T.H. analyzed the results. A.C.C., K.K.L., M.E.L., S.-A.D., M.F., and T.H wrote the manuscript. All authors reviewed the manuscript.

## Competing interests

The authors declare no competing interests.
