## [Peer Review File · Nature Communications]

Reviewers' comments:

Reviewer #1 (Remarks to the Author):

Christensen et al. demonstrate in their study that parvalbumin-related perineuronal nets (PNNs) support the stability of the grid cell network. That was done by disrupting PNNs in the MEC of adult rats using chABC, and recording from the rats during exploration of familiar and novel environments. In addition to loss of stability, various parameters such as out-of-field firing and spatial information changed as well. To check the validity of their prediction on the grid cells, the authors also performed a simulation with altered synaptic weights that emphasized the reduced inhibition during the disruption of the PNNs. They furthermore recorded place cells from CA1 area from rats with and without perineuronal nets, and showed the reduction of stability also in the hippocampus.

The paper is written very clearly, and the results are both interesting and important.

We have a few questions:

1. How did the removal of PNNs affect other types of cells apart from grid cells in the MEC?
2. Figures 2 and 7 – could the differences between the control and chABC groups be explained by the lower firing rates in the chABC rats? We suggest a procedure in which the control group spikes are deleted randomly to fit the rates in the chABC group, and then see how that affects all of the parameters.
3. Figure 3 – please add an analysis of the phase shift development and the changes in grid spacing across the different conditions.
4. "Contrary to experimental data, we found increased peak rates of grid fields in the reduced-inhibition model". As the authors discuss, they did manage to get this effect in a simulation of an "increased capacitance model" – we ask that they display the parameters of this model too in the Methods and its output as a supplementary figure.
5. "Reducing excitatory to inhibitory synaptic strength mimics the effect of PNN removal in a simulated grid cell network" – The authors should reserve this, because their simulation does not explain their main result of reduced stability of the grids.
6. Figure 3 c-d: Only units that were defined as grid cells in all the sessions were used. How many cells lost their spatial tuning in the chABC vs. the control groups (between the familiar and novel arenas)?

Reviewer #2 (Remarks to the Author):

This paper examined the role of perineuronal nets (PNNs) in spacial memory conferred by grid cells. These cells work hand in glove with place cells in the hippocampus to map an animals spacial environment. Grid cells have been previously shown to encode a less plastic environment. In light of the previously described roles of PNNs in stabilizing synapses the authors ventures to examine the hypothesis that PNNs are involved in grid cell network stability. The primary approach is the commonly used enzymatic digestion using chondroitinase ABC while assessing neuronal network activity through extracellular unit recording. The authors did an extensive set of studies that collectively support the following conclusions: PNNs on cortical PV neurons that project to the hippocampus maintain the functional organization of microcircuits involved in spatial memory. Without PNNs grid cells could not establish a representation of a novel environment and destabilizes hippocampal place cells.

These findings build logically on previously described or hypothesized functions of PNNs and make an important contribution to our understanding of spacial memories. They are, however, somewhat descriptive as they approach the experiments through the "block" experiment. There are no experiments that show encoding of new information by alterations of PNNs. They don't

provide mechanistic insights to how PNNs are formed, by which cells and how dynamic PNNs are in normal biology. They only provide a somewhat binary answer that without PNNs new memories don't form.

Nevertheless the experiments are solid and by and large support the conclusions.

Reviewer #3 (Remarks to the Author):

Christiansen et al found that degradation of perineural nets (PNN) in the medial entorhinal cortex (MEC) induces :

- 1- lower firing rate in both putative interneuron and principal cells
- 2- reduced spatial specificity and stability of grid cells, particularly in a novel environment
- 3- increased power and speed correlation but reduced frequency of theta oscillations
- 4- decreased stability of place cells

Although I find this article and the rationale interesting I have some concerns :

1- I find the description of the protocol not very clear. Similarly, it is not clear whether the total number of grid cells recorded (reported on page 5) was also used in the different analysis. The authors recorded MEC cells from 14 rats using a protocol of 5 successive sessions. If I understood correctly (from the method section), each rat was submitted only to three days of recording (from the moment grid cells were identified) : the first day included the 5 sessions protocol (fam1, nov 1-2-3, fam2), and the two following days included only 1 session in the previously novel environment . All the data reported are from day 1 recordings. Day 2 recordings were used to assess grid cell stability in the novel environment, whereas recordings from day 3 were not used. Since the total number of grid cells were 86 in control rats and 63 in the chABC group, then on average the authors were able to record roughly 10 simultaneous grid cells from each rat. However, if this was the case, the number of grid cells pairs should be higher than that reported. To what exactly correspond the number of data points reported in figure 3 e and g ?

2- I find the discussion of the results quite limited in some parts. In particular when the results are not fully consistent with the literature (or with other results from the same study), the authors do not discuss such discrepancy. For example, the major interpretation that the authors report is that PNN degradation affect mainly PV interneurons, and this in turn influence grid cell spatial properties (mainly stability). But the results they obtain are not fully consistent with those reported by Miao et al (2017), that directly tested the effects induced by pharmacogenetic inactivation of PV cells on grid cell firing properties, and yet no discussion is provided (except for a small paragraph reporting such discrepancy). Similarly, the authors showed opposite effects on theta oscillations from MEC and hippocampus following PNN degradation, but I couldn't find any explanation for that.

3- What is the effect of PNN disruption on the other MEC spatial selective cells ? Are speed cells affected ?

4- I find quite unexpected that chABC-treated rats show reduced spatial stability also in the familiar environment but only following exposure to the novel environment. Does this mean that rats do not « recognize » the familiar environment ? Is there any changes in their behavior ?

5- It would be useful to indicate the number of grid cells recorded from each rat, and to show some LFP traces (raw and filtered) from both MEC and hippocampus

We thank the reviewers for their insightful comments and suggestions to improve our manuscript. We have performed the analyses requested and made changes to parts of the manuscript accordingly. In detail, we have performed several new analyses on animal behavior, effects on speed-coding and head-direction cells, and details on the effects on grid cell and network coding. Our new analyses do not alter the major conclusions of the paper, but rather add to it. During the revision we have made substantial additions to the discussion regarding the effects of using chABC, network plasticity and the grid cell modeling. We have also added all statistical test parameters and results from main figures as supplementary tables 2-10.

We believe that the manuscript is substantially improved.

In the manuscript, new text is highlighted in blue, while deleted text is highlighted in red.

Reviewers' comments:

Reviewer #1 (Remarks to the Author):

Christensen et al. demonstrate in their study that parvalbumin-related perineuronal nets (PNNs) support the stability of the grid cell network. That was done by disrupting PNNs in the MEC of adult rats using chABC, and recording from the rats during exploration of familiar and novel environments. In addition to loss of stability, various parameters such as out-of-field firing and spatial information changed as well. To check the validity of their prediction on the grid cells, the authors also performed a simulation with altered synaptic weights that emphasized the reduced inhibition during the disruption of the PNNs. They furthermore recorded place cells from CA1 area from rats with and without perineuronal nets, and showed the reduction of stability also in the hippocampus.

The paper is written very clearly, and the results are both interesting and important.

We have a few questions:

1. How did the removal of PNNs affect other types of cells apart from grid cells in the MEC?

The analysis for head-direction cells and speed cells are now included in the result chapter and as Supplementary Figure S6 and S7. Both grid, head direction and speed cells in the chABC treated group show reduced burst activity and reduced spiking variability. We observed a slight reduction in average firing rate for broad-spiking units, but no significant changes in average firing rate for grid cells, head-direction cells or speed-modulated cells.

2. Figures 2 and 7 – could the differences between the control and chABC groups be explained by the lower firing rates in the chABC rats? We suggest a procedure in which the control group spikes are deleted randomly to fit the rates in the chABC group, and then see how that affects all of the parameters.

The previous reported difference in firing rate between control and chABC was calculated for all broad-spiking units. We re-examined the analysis and found that there is no difference in average firing rates for grid cells between the groups, thus the downsampling was not necessary. The significant changes in spatial information and ISI CV are not dependent on average firing rate.

3. *Figure 3 – please add an analysis of the phase shift development and the changes in grid spacing across the different conditions.*

We have performed similar analysis as Barry et al., (2012) to investigate the effect of novel environment on grid spacing and found increased spacing in the novel environment in both groups (Supplementary Figure S9). While our changes in spacing was not as pronounced as those of Barry et al., this is likely because the environments used by Barry et al. differed in texture, odour and visual appearance, while we used similar boxes in different rooms.

4. *“Contrary to experimental data, we found increased peak rates of grid fields in the reduced-inhibition model”. As the authors discuss, they did manage to get this effect in a simulation of an “increased capacitance model” – we ask that they display the parameters of this model too in the Methods and its output as a supplementary figure.*

We have included a table with the parameters in the method section and the output from the increased capacitance model in Supplementary Figure S12. We also added further to the discussion about the simulated data.

5. *“Reducing excitatory to inhibitory synaptic strength mimics the effect of PNN removal in a simulated grid cell network” – The authors should reserve this, because their simulation does not explain their main result of reduced stability of the grids.*

We agree and have rephrased the statement to: *Reducing excitatory to inhibitory synaptic strength in a simulated grid cell network mimics the effect of PNN removal in a **familiar environment**.*

6. *Figure 3 c-d: Only units that were defined as grid cells in all the sessions were used. How many cells lost their spatial tuning in the chABC vs. the control groups (between the familiar and novel arenas)?*

We have now rephrased the main text and figure legend to make it clear which cells were included in the analysis. Many grid cells fall below threshold for gridness score in the first session in the novel environment, but reappear as grid cells after further exposure to the novel environment. The cells we included in the analysis of grid cells across the novel environment experiment were therefore required to:

- 1) have a gridness score above threshold in Familiar I
- 2) units could be identified in all five recording sessions.

Unfortunately gridness score is a somewhat artificial measure and it is not suited to capture the dynamic changes in spatial representations in a novel environment. We rather

emphasize the spatial information and stability in the novel environment. However, both groups show a significant reduction in gridness score when going from familiar 1 to novel 1. Although the grid fields get more clear in novel 2 and 3 (and the gridness scores increase), only 38% of the cells above threshold in Familiar 1 is above threshold in Novel 3, and we did not see a significant difference between grid cells recorded from controls and chABC treated rats (37 and 43% of grid cells from Familiar 1 above threshold in Novel 3, respectively).

Reviewer #2 (Remarks to the Author):

This paper examined the role of perineuronal nets (PNNs) in spacial memory conferred by grid cells. These cells work hand in glove with place cells in the hippocampus to map an animals spacial environment. Grid cells have been previously shown to encode a less plastic environment. In light of the previously described roles of PNNs in stabilizing synapses the authors ventures to examine the hypothesis that PNNs are involved in grid cell network stability. The primary approach is the commonly used enzymatic digestion using chondroitinase ABC while assessing neuronal network activity through extracellular unit recording. The authors did an extensive set of studies that collectively support the following conclusions: PNNs on cortical PV neurons that project to the hippocampus maintain the functional organization of microcircuits involved in spatial memory. Without PNNs grid cells could not establish a representation of a novel environment and destabilizes hippocampal place cells.

These findings build logically on previously described or hypothesized functions of PNNs and make an important contribution to our understanding of spacial memories. They are, however, somewhat descriptive as they approach the experiments through the "block" experiment. There are no experiments that show encoding of new information by alterations of PNNs. They don't provide mechanistic insights to how PNNs are formed, by which cells and how dynamic PNNs are in normal biology. They only provide a somewhat binary answer that without PNNs new memories don't form.

Nevertheless the experiments are solid and by and large support the conclusions.

We thank the reviewer for raising an important question and we agree that there is a general lack of knowledge about the mechanism of PNN remodelling, and our study did not address this question. Targeted manipulations of single PNN components or the specific cell types would be highly beneficial for understanding the role of PNNs at a detailed level.

Unfortunately, the tools needed for these kinds of experiments are limited. For example, inhibitors of matrix regulating enzymes e.g. metalloproteinases are mostly non-specific. Furthermore, transgenic knock out of PNN proteins have either limited effects on the overall structure or cause gradual loss of PNN components over time and are thus prone to compensatory effects (e.g. Favuzzi et al., 2017 *Neuron*; Rowlands et al., 2018 *JNeurosci*). Since the impact of PNNs on the grid cell network had not previously been studied we chose enzymatic digestion which has been used in many brain areas previously. Using

chondroitinase, the PNN structure decays rapidly and we aimed to study how a network that has undergone normal development processes information after a sudden loss of PNNs.

Reviewer #3 (Remarks to the Author):

Christiansen et al found that degradation of perineural nets (PNN) in the medial entorhinal cortex (MEC) induces :

- 1- lower firing rate in both putative interneuron and principal cells*
- 2- reduced spatial specificity and stability of grid cells, particularly in a novel environment*
- 3- increased power and speed correlation bur reduced frequency of theta oscillations*
- 4- decreased stability of place cells*

Although I find this article and the rationale interesting I have some concerns :

1- I find the description of the protocol not very clear. Similarly, it is not clear whether the total number of grid cells recorded (reported on page 5) was also used in the different analysis.

The authors recorded MEC cells from 14 rats using a protocol of 5 successive sessions. If I understood correctly (from the method section), each rat was submitted only to three days of recording (from the moment grid cells were identified) : the first day included the 5 sessions protocol (fam1, nov 1-2-3, fam2), and the two following days included only 1 session in the previously novel environment . All the data reported are from day 1 recordings. Day 2 recordings were used to asses grid cell stability in the novel environment, whereas recordings from day 3 were not used. Since the total number of grid cells were 86 in control rats and 63 in the chABC group, then on average the authors were able to record roughly 10 simultaneous grid cells from each rat. However, if this was the case, the number of grid cells pairs should be higher that that reported. To what exactly correspond the number of data points reported in figure 3 e and g ?

We thank the referee for pointing out this problem and have revised both the methods and results sections to make it clear which units are included in the different analyses. In the result section we now emphasize the difference between familiar and novel environment recordings and clarify the protocol used for sampling data for the different analyses. We have also updated the method section to make the distinction between protocols more clear. We hope this clarifies which units are included in the different analyses.

2- I find the discussion of the results quite limited in some parts. In particular when the results are not fully consistent with the litarure (or with other results from the same study), the authors do not discuss such discrepancy. For exemple, the major interpretation that the authors report is that PNN degradation affect mainly PV interneurons, and this in turn influence grid cell spatial properties (mainly stability). But the results they obtain are not fully consistent with those reported by Miao et al (2017), that directly tested the effects induced by pharmacogenetic inactivation of PV cells on grid cell firing properties, and yet no

discussion is provided (except for a small paragraph reporting such discrepancy).

We have expanded the discussion about this important issue pointed out by the referee. While we believe that the direct effect of PNN degradation is mainly confined to PV cells and synapses onto PV cells, the indirect effects are more complex. Although the reduced spiking of PV cells can cause a reduction in spatial specificity (as shown by our simulations and the results by Miao et al.), it remains unknown if and how reduced PV activity would affect representations in a novel environment. For the familiar environment both our study and the study by Miao and colleagues found reduced spatial information and specificity. In addition we found reduced maximum firing rate while Miao did not. The most apparent discrepancy however, is the reduction in spatial stability observed by Miao et al during PV silencing. The perturbation of PV cell activity performed by Miao et al cause a substantial transient reduction in PV cell activity and it is therefore be difficult to compare their results directly with our data. Since PV interneurons are connected to most cell types in MEC (Buetfering et al., 2014), it is unknown if there are indirect effects of the chemogenetic perturbation. Interestingly, optogenetic activation of PV neurons did not alter spatial specificity of grid cells, although they were robustly inhibited (Buetfering et al., 2014). This points to a strong attractor dynamics within the grid cell network that it is resilient to a range of perturbations. Since we only observed a change in spatial stability during and after the network encodes new information, we find it unlikely that activity levels of PV cells is sufficient to explain our results from novel environment experiments.

Similarly, the authors showed opposite effects on theta oscillations from MEC and hippocampus following PNN degradation, but I couldn't find any explanation for that.

While theta power score is similar in MEC and hippocampus, the frequency score is increased in the chABC group in MEC and reduced in hippocampus. Unfortunately, we do not have an explanation for that phenomenon. Since theta in MEC and HPC is strongly connected, it could be that the slight change of theta in MEC alters how theta is modulated by the running speed in hippocampus.

3- What is the effect of PNN disruption on the other MEC spatial selective cells ? Are speed cells affected ?

We have now included the analysis of head direction cells and speed-modulation (Supplementary Figure S6, S7). It is mainly the temporal firing statistics that is altered, by reduced bursting and spiking variability in both grid, speed and head-direction cells.

4- I find quite unexpected that chABC-treated rats show reduced spatial stability also in the familiar environment but only following exposure to the novel environment. Does this mean that rats do not « recognize » the familiar environment ? Is there any changes in their behavior ?

We have included more behavioral analysis and expanded the discussion around this intriguing finding in the revised manuscript. It is hard to know how the rats perceived the familiar environment after being exposed to the novel environment, but both behavioral and electrophysiological data indicate that they did not experience it as a novel environment for several reasons: First, they returned to the original familiar grid map (although weakened). Second, the novelty signal in the theta did not appear in “familiar 2”. Finally, the behavioral data indicate that the two groups of animals experienced it as a familiar environment since the running path and running speed of the animals were similar. This has been added to Supplementary Figure S8.

5- It would be useful to indicate the number of grid cells recorded from each rat, and to show some LFP traces (raw and filtered) from both MEC and hippocampus

We have included a table with summary of all grid cells recorded from the different rats (Supplementary table 1) and a Supplementary Figure (S10) with LFP traces.

REVIEWERS' COMMENTS:

Reviewer #1 (Remarks to the Author):

I am satisfied with the answers to the comments in the rebuttal, and the changes in the manuscript.

In my view the paper is now fit for publication.

Reviewer #2 (Remarks to the Author):

This is a revised paper examining the role of perineuronal nets (PNNs) on the stability of the grid-cell network underlying spatial learning. The authors primarily show that disruption of PNNs using the enzyme chondroitinase ABC impairs spatial maps and impairs ability to acquire a new map. I previously emphasized that this study while descriptive is important and consistent with prior findings regarding the stability of synaptic networks, hence these findings are not unexpected. Given that they are novel they deserve to be published. Comments provided by other reviewers appear to have been addressed extensively. Future studies must look in greater detail as to whether specific synapses are destabilized and whether intrinsic electrical properties may have been altered, as this has been suggested by others, i.e. Tewari 2018 as cited in the manuscript.

Reviewer #3 (Remarks to the Author):

I carefully read the revised version of the manuscript and in my opinion the authors addressed correctly all raised questions.